# Degradation-Aware Unfolding Half-Shuffle Transformer for Spectral Compressive Imaging

**Yuanhao Cai** [1,2,*], **Jing Lin** [1,2,*], **Haoqian Wang** [1,2,†], **Xin Yuan** [3],
**Henghui Ding** [4], **Yulun Zhang** [4], **Radu Timofte** [4,5], **Luc Van Gool** [4]
[1] Shenzhen International Graduate School, Tsinghua University,
[2] Shenzhen Institute of Future Media Technology,
[3] Westlake University, [4] ETH Zürich, [5] University of Würzburg

## Abstract

In coded aperture snapshot spectral compressive imaging (CASSI) systems, hyperspectral image (HSI) reconstruction methods are employed to recover the spatial-spectral signal from a compressed measurement. Among these algorithms, deep unfolding methods demonstrate promising performance but suffer from two issues. Firstly, they do not estimate the degradation patterns and ill-posedness degree from CASSI to guide the iterative learning. Secondly, they are mainly CNN-based, showing limitations in capturing long-range dependencies. In this paper, we propose a principled Degradation-Aware Unfolding Framework (DAUF) that estimates parameters from the compressed image and physical mask, and then uses these parameters to control each iteration. Moreover, we customize a novel Half-Shuffle Transformer (HST) that simultaneously captures local contents and non-local dependencies. By plugging HST into DAUF, we establish the first Transformer-based deep unfolding method, Degradation-Aware Unfolding Half-Shuffle Transformer (DAUHST), for HSI reconstruction. Experiments show that DAUHST surpasses state-of-the-art methods while requiring cheaper computational and memory costs. Code and models are publicly available at https://github.com/caiyuanhao1998/MST

## 1 Introduction

Hyperspectral images (HSIs) have more spectral bands than normal RGB images to store more detailed information. Thus, HSIs are widely applied in image recognition [1, 2, 3], object detection [4, 5, 6], tracking [7, 8, 9], medical image processing [10, 11, 12], remote sensing [13, 14, 15, 16], *etc*. To obtain HSIs, traditional imaging systems use spectrometers to scan the scenes along the spectral or spatial dimensions, usually requiring a long time. These imaging systems fail to capture dynamic objects. Recently, snapshot compressive imaging (SCI) systems [17, 18, 19] have been developed to capture HSIs at video rate. Among these SCI systems, coded aperture snapshot spectral imaging (CASSI) [17, 20, 21] stands out for its impressive performance. CASSI uses a coded aperture and a disperser to modulate the HSI signal at different wavelengths, and then mixes all modulated signal to generate a 2D compressed

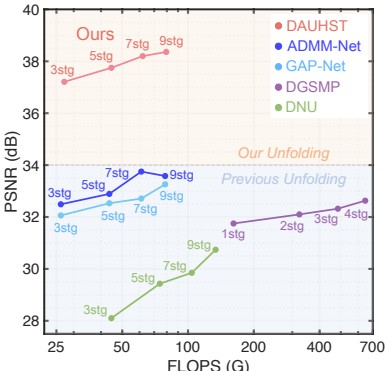

Figure 1: PSNR-FLOPS comparisons of DAUHST and SOTA unfolding methods.

measurement. Subsequently, HSI restoration methods are employed to solve the CASSI inverse problem, *i.e.*, restore the HSIs from the measurement. These methods are divided into four categories.

**(i)** Model-based methods [17, 22, 23, 24, 25, 26, 27, 28] rely on hand-crafted image priors, *e.g.*, total variation [26], sparsity [17, 22], low-rank [23], *etc*. These methods have theoretically proven

---

*Equal Contribution, † Corresponding Author

36th Conference on Neural Information Processing Systems (NeurIPS 2022).

properties and can be interpreted. Yet, these methods need manual parameter tweaking, which slows down reconstruction. Also, they suffer from limited representation capacity and generalization ability.

**(ii)** Plug-and-play (PnP) algorithms [29, 30, 31] plug pre-trained denoising networks into traditional model-based methods to solve the HSI reconstruction problem. Nonetheless, the pre-trained networks in PnP methods are fixed without re-training, therefore limiting the performance.

**(iii)** End-to-end (E2E) algorithms employ a powerful model, usually a convolutional neural network (CNN) [12, 20, 32, 33], to learn the E2E mapping function from a measurement to the desired HSIs. E2E methods enjoy the power of deep learning. However, they learn a brute-force mapping from the compressed measurement to the underlying spectral images, thereby ignoring the working principles of CASSI systems. They come without theoretically proven properties, interpretability, and flexibility because the imaging models widely differ from each other for various hardware systems.

**(iv)** Deep unfolding methods [34, 35, 36, 37, 38, 39] adopt a multi-stage network to map the measurement into the HSI cube. Each stage usually includes two phases, *i.e.*, linear projection followed by passing the signal through a single-stage network that learns the underlying denoiser prior. In deep unfolding methods, the network architecture is intuitively interpretable by explicitly characterizing the image priors and the system imaging model. Besides, these methods also enjoy the power of deep learning and thus have great potential. Yet, this potential has not been fully explored.

Existing deep unfolding algorithms suffer from two issues. **(a)** The iterative learning is highly related to the CASSI system. However, current unfolding methods do not estimate CASSI degradation patterns and ill-posedness degree to adjust the linear projection and denoising network in each iteration. **(b)** Existing deep unfolding methods are mainly CNN-based, therefore showing limitations in capturing non-local self-similarity and long-range dependencies, both critical for HSI reconstruction.

Recently, the emerging Transformer [40] has provided a solution to tackle the drawbacks of CNN. Due to its strong capability in modeling the interactions of non-local spatial regions, Transformer has been widely applied in image classification [41, 42, 43], object detection [44, 45, 46], semantic segmentation [47, 48, 49], human pose estimation [50, 51, 52], image restoration [53, 54, 55], *etc*. Yet, the use of Transformer is confronted with two main issues. **(a)** The computational complexity of global Transformer [42] is quadratic to the spatial dimensions. This cost is sometimes unaffordable. **(b)** The receptive fields of local Transformer [41] are limited within position-specific windows. Thus, some tokens with highly-related contents can not match each other when computing self-attention.

To address the above problems, in this paper, we firstly formulate a principled Degradation-Aware Unfolding Framework (DAUF) based on maximum *a posteriori* (MAP) theory for HSI reconstruction. Different from previous deep unfolding methods, our DAUF implicitly estimates informative parameters from the degraded compressed measurement and the physical mask used in the modulation. Then DAUF feeds the parameters, which capture key cues of CASSI degradation patterns and ill-posedness degree, into each iteration to adaptively scale the linear projection and provide the noise level information for the denoising network. Secondly, we design a novel Half-Shuffle Transformer (HST) as the denoiser prior in each iteration. Our HST can jointly extract local contextual information and model non-local dependencies, while requiring much cheaper computational costs than global Transformer. We achieve this by customizing a Half-Shuffle Multi-head Self-Attention (HS-MSA) mechanism that composes the basic unit of HST. More specifically, our HS-MSA has two branches, *i.e.*, *local branch* and *non-local branch*. The *local branch* calculates the self-attention within the local window while the *non-local branch* shuffles the tokens and captures cross-window interactions. We plug HST into DAUF to establish an iterative architecture, Degradation-Aware Unfolding Half-Shuffle Transformer (DAUHST). With the proposed techniques, DAUHST models dramatically outperform state-of-the-art (SOTA) deep unfolding methods with the same number of stages by **over 4 dB**, as shown in Fig. 1.

In a nutshell, our contributions can be summarized as follows:

**(i)** We formulate a principled MAP-based unfolding framework DAUF for HSI reconstruction.

**(ii)** We propose a novel Transformer HST and plug it into DAUF to establish DAUHST. To the best of our knowledge, DAUHST is the first Transformer-based deep unfolding method for HSI restoration.

**(iii)** DAUHST outperforms SOTA methods by a large margin while requiring cheaper computational and memory costs. Besides, DAUHST yields more visually pleasant results in real HSI reconstruction.

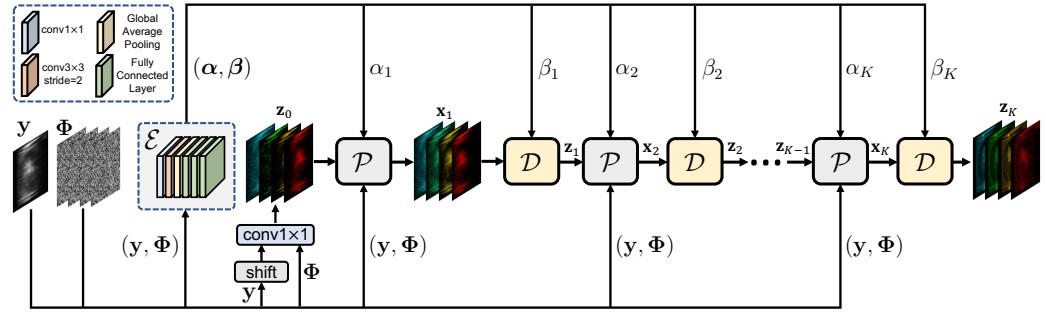

Figure 2: The architecture of our DAUF with $K$ stages (iterations). $\mathcal{E}$ estimates informative parameters from the compressed measurement $\mathbf{y}$ and sensing matrix $\mathbf{\Phi}$. The estimated parameters $\boldsymbol{\alpha}$ and $\boldsymbol{\beta}$ are fed into each stage of subsequent iterative learning. $\mathcal{P}$ and $\mathcal{D}$ denote the linear projection and denoising network in each stage.

## 2 Proposed Method

### 2.1 Degradation Model of CASSI

In CASSI, we denote the vectorized measurement as $\mathbf{y} \in \mathbb{R}^n$, where $n = H(W + d(N_\lambda - 1))$. $H$, $W$, $d$, and $N_\lambda$ denote the HSI's height, width, shifting step in dispersion, and total number of wavelengths. Given the vectorized shifted HSI signal $\mathbf{x} \in \mathbb{R}^{nN_\lambda}$ and the sensing matrix $\mathbf{\Phi} \in \mathbb{R}^{n \times nN_\lambda}$ that is determined by the physical mask, the degradation model of CASSI can be formulated as

$$\mathbf{y} = \mathbf{\Phi}\mathbf{x} + \mathbf{n}, \tag{1}$$

where $\mathbf{n} \in \mathbb{R}^n$ represents the vectorized imaging noise on the measurement. As analyzed in [56, 57, 58], $\mathbf{\Phi}$ is a fat, sparse, and highly structured matrix that is hard to handle. Please refer to the supplementary material for details about the mathematical model of CASSI. Then the task of HSI reconstruction is given $\mathbf{y}$ (captured by the camera) and $\mathbf{\Phi}$ (calibrated based on pre-design), solving $\mathbf{x}$.

### 2.2 Degradation-Aware Unfolding Framework

Previous unfolding frameworks [34, 35, 36, 37] do not estimate the CASSI degradation patterns to adjust the iterative learning. To alleviate this limitation, we formulate a principled Degradation-Aware Unfolding Framework (DAUF) as depicted in Fig. 2. DAUF starts from the MAP theory. In particular, the original HSI signal could be estimated by minimizing the following energy function as

$$\hat{\mathbf{x}} = \arg\min_{\mathbf{x}} \frac{1}{2}||\mathbf{y} - \mathbf{\Phi}\mathbf{x}||^2 + \tau R(\mathbf{x}), \tag{2}$$

where $\frac{1}{2}||\mathbf{y} - \mathbf{\Phi}\mathbf{x}||^2$ is the data fidelity term, $R(\mathbf{x})$ is the image prior term, and $\tau$ is a hyperparameter balancing the importance. By introducing an auxiliary variable $\mathbf{z}$, Eq. (2) can be reformulated as

$$\hat{\mathbf{x}} = \arg\min_{\mathbf{x}} \frac{1}{2}||\mathbf{y} - \mathbf{\Phi}\mathbf{x}||^2 + \tau R(\mathbf{z}), \quad s.t. \ \mathbf{z} = \mathbf{x}. \tag{3}$$

This is a constrained optimization problem. To obtain an unfolding inference, we adopt half-quadratic splitting (HQS) algorithm for its simplicity and fast convergence. Then Eq. (3) is solved by minimizing

$$\mathcal{L}_\mu(\mathbf{x}, \mathbf{z}) = \frac{1}{2}||\mathbf{y} - \mathbf{\Phi}\mathbf{x}||^2 + \tau R(\mathbf{z}) + \frac{\mu}{2}||\mathbf{z} - \mathbf{x}||^2, \tag{4}$$

where $\mu$ is a penalty parameter that forces $\mathbf{x}$ and $\mathbf{z}$ to approach the same fixed point. Subsequently, Eq. (4) can be solved by decoupling $\mathbf{x}$ and $\mathbf{z}$ into the following two iterative sub-problems as

$$\mathbf{x}_{k+1} = \arg\min_{\mathbf{x}} ||\mathbf{y} - \mathbf{\Phi}\mathbf{x}||^2 + \mu||\mathbf{x} - \mathbf{z}_k||^2, \quad \mathbf{z}_{k+1} = \arg\min_{\mathbf{z}} \frac{\mu}{2}||\mathbf{z} - \mathbf{x}_{k+1}||^2 + \tau R(\mathbf{z}), \tag{5}$$

where $k = 0, 1, \ldots, K-1$ indexes the iteration. Note that the data fidelity term is associated with a quadratic regularized least-squares problem, i.e., $\mathbf{x}_{k+1}$ in Eq. (5). It has a closed-form solution as

$$\mathbf{x}_{k+1} = (\mathbf{\Phi}^\mathsf{T}\mathbf{\Phi} + \mu\mathbf{I})^{-1}(\mathbf{\Phi}^\mathsf{T}\mathbf{y} + \mu\mathbf{z}_k), \tag{6}$$

where $\mathbf{I}$ is an identity matrix. Since $\mathbf{\Phi}$ is a fat matrix, $(\mathbf{\Phi}^\mathsf{T}\mathbf{\Phi} + \mu\mathbf{I})$ will be large and thus we simplify the computation of the inverse problem $(\mathbf{\Phi}^\mathsf{T}\mathbf{\Phi} + \mu\mathbf{I})^{-1}$ by the matrix inversion formula as

$$(\mathbf{\Phi}^\mathsf{T}\mathbf{\Phi} + \mu\mathbf{I})^{-1} = \mu^{-1}\mathbf{I} - \mu^{-1}\mathbf{\Phi}^\mathsf{T}(\mathbf{I} + \mathbf{\Phi}\mu^{-1}\mathbf{\Phi}^\mathsf{T})^{-1}\mathbf{\Phi}\mu^{-1}. \tag{7}$$

By plugging Eq. (7) into Eq. (6), we can reformulate Eq. (6) as

$$\mathbf{x}_{k+1} = \frac{\mathbf{\Phi}^\mathsf{T}\mathbf{y} + \mu\mathbf{z}_k}{\mu} - \frac{\mathbf{\Phi}^\mathsf{T}(\mathbf{I} + \mathbf{\Phi}\mu^{-1}\mathbf{\Phi}^\mathsf{T})^{-1}\mathbf{\Phi}\mathbf{\Phi}^\mathsf{T}\mathbf{y}}{\mu^2} - \frac{\mathbf{\Phi}^\mathsf{T}(\mathbf{I} + \mathbf{\Phi}\mu^{-1}\mathbf{\Phi}^\mathsf{T})^{-1}\mathbf{\Phi}\mathbf{z}_k}{\mu}. \tag{8}$$

In CASSI systems, $\mathbf{\Phi}\mathbf{\Phi}^\mathsf{T}$ is a diagonal matrix which can be defined as $\mathbf{\Phi}\mathbf{\Phi}^\mathsf{T} \overset{\text{def}}{=} \mathrm{diag}\{\psi_1, \dots, \psi_n\}$. By plugging $\mathbf{\Phi}\mathbf{\Phi}^\mathsf{T}$ into $(\mathbf{I} + \mathbf{\Phi}\mu^{-1}\mathbf{\Phi}^\mathsf{T})^{-1}$ and $(\mathbf{I} + \mathbf{\Phi}\mu^{-1}\mathbf{\Phi}^\mathsf{T})^{-1}\mathbf{\Phi}\mathbf{\Phi}^\mathsf{T}$, we obtain:

$$
\begin{aligned}
(\mathbf{I} + \mathbf{\Phi}\mu^{-1}\mathbf{\Phi}^\mathsf{T})^{-1} &= \mathrm{diag}\Big\{\frac{\mu}{\mu + \psi_1}, \dots, \frac{\mu}{\mu + \psi_n}\Big\}, \\
(\mathbf{I} + \mathbf{\Phi}\mu^{-1}\mathbf{\Phi}^\mathsf{T})^{-1}\mathbf{\Phi}\mathbf{\Phi}^\mathsf{T} &= \mathrm{diag}\Big\{\frac{\mu\psi_1}{\mu + \psi_1}, \dots, \frac{\mu\psi_n}{\mu + \psi_n}\Big\}.
\end{aligned}
\tag{9}
$$

Let $\mathbf{y} \overset{\text{def}}{=} [y_1, \dots, y_n]^\mathsf{T}$ and $[\mathbf{\Phi}\mathbf{z}_k]_i$ denotes the $i$-th element of $\mathbf{\Phi}\mathbf{z}_k$. We plug Eq. (9) into Eq. (8) as

$$
\begin{aligned}
\mathbf{x}_{k+1} &= \frac{\mathbf{\Phi}^\mathsf{T}\mathbf{y}}{\mu} + \mathbf{z}_k - \frac{1}{\mu}\mathbf{\Phi}^\mathsf{T}\Big[\frac{y_1\psi_1 + \mu[\mathbf{\Phi}\mathbf{z}_k]_1}{\mu + \psi_1}, \dots, \frac{y_n\psi_n + \mu[\mathbf{\Phi}\mathbf{z}_k]_n}{\mu + \psi_n}\Big]^\mathsf{T} \\
&= \mathbf{z}_k + \mathbf{\Phi}^\mathsf{T}\Big[\frac{y_1 - [\mathbf{\Phi}\mathbf{z}_k]_1}{\mu + \psi_1}, \dots, \frac{y_n - [\mathbf{\Phi}\mathbf{z}_k]_n}{\mu + \psi_n}\Big]^\mathsf{T}.
\end{aligned}
\tag{10}
$$

Note that $\{y_i - [\mathbf{\Phi}\mathbf{z}_k]_i\}_{i=1}^n$ can be directly updated by $\mathbf{y} - \mathbf{\Phi}\mathbf{z}_k$, and $\{\psi_i\}_{i=1}^n$ is pre-calculated and stored in $\mathbf{\Phi}\mathbf{\Phi}^\mathsf{T}$. Thus, by element-wise computation in Eq. (10), $\mathbf{x}_{k+1}$ can be updated very efficiently. According to Eq. (5), the penalty parameter $\mu$ should be large enough so that $\mathbf{x}$ and $\mathbf{z}$ can approach approximately the same fixed point. This indicates that $\mu$ controls the convergence and output of each iteration. Thus, instead of manually tweaking $\mu$, we set $\mu$ as a series of iteration-specific parameters to be automatically estimated from the CASSI system. We denote $\mu$ in the $k$-th iteration as $\mu_k$.

Returning to Eq. (5), we also set $\tau$ as iteration-specific parameters and $\mathbf{z}_{k+1}$ can be reformulated as

$$\mathbf{z}_{k+1} = \arg\min_{\mathbf{z}} \ \frac{1}{2(\sqrt{\tau_{k+1}/\mu_{k+1}})^2} \, ||\mathbf{z} - \mathbf{x}_{k+1}||^2 + R(\mathbf{z}). \tag{11}$$

From the perspective of Bayesian probability, Eq. (11) is equivalent to denoising image $\mathbf{x}_{k+1}$ with a Gaussian noise at level $\sqrt{\tau_{k+1}/\mu_{k+1}}$ [29]. To conveniently solve Eq. (11), we set $\frac{1}{(\sqrt{\tau_{k+1}/\mu_{k+1}})^2} = \mu_{k+1}/\tau_{k+1}$ as parameters to be estimated from CASSI. Let $\alpha_k \overset{\text{def}}{=} \mu_k$, $\boldsymbol{\alpha} \overset{\text{def}}{=} [\alpha_1, \dots, \alpha_K]$, $\beta_k \overset{\text{def}}{=} \mu_k/\tau_k$, and $\boldsymbol{\beta} \overset{\text{def}}{=} [\beta_1, \dots, \beta_K]$. Then we can formulate our DAUF as an iterative scheme:

$$(\boldsymbol{\alpha}, \boldsymbol{\beta}) = \mathcal{E}(\mathbf{y}, \mathbf{\Phi}), \quad \mathbf{x}_{k+1} = \mathcal{P}(\mathbf{y}, \mathbf{z}_k, \alpha_{k+1}, \mathbf{\Phi}), \quad \mathbf{z}_{k+1} = \mathcal{D}(\mathbf{x}_{k+1}, \beta_{k+1}), \tag{12}$$

where $\mathcal{E}$ denotes the parameter estimator that takes the compressed measurement $\mathbf{y}$ and the sensing matrix $\mathbf{\Phi}$ of the CASSI system as inputs, $\mathcal{P}$ equivalent to Eq. (10) denotes the linear projection, and $\mathcal{D}$ represents the Gaussian denoiser solving Eq. (11). $\mathbf{z}_0$ is initialized by passing the shifted $\mathbf{y}$ concatenated with $\mathbf{\Phi}$ through a $conv1 \times 1$ (convolution with $1 \times 1$ kernel). Fig. 2 shows the architecture of $\mathcal{E}$. It consists of a $conv1 \times 1$, a strided $conv3 \times 3$, a global average pooling, and three fully connected layers. Through $\mathcal{E}$, DAUF captures critical cues from CASSI by learning the degradation patterns and ill-posedness degree caused by the mask-modulation and dispersion-integration. Parameters $\boldsymbol{\alpha}$ and $\boldsymbol{\beta}$ estimated by $\mathcal{E}$ direct the iterative learning by adaptively scaling the linear projection in Eq. (10) and providing noise level information for the denoiser prior in Eq. (11).

### 2.3 Half-Shuffle Transformer

When designing the denoiser prior, previous deep unfolding methods [34, 35, 36, 37] mainly adopt CNNs, showing limitations in capturing long-range dependencies. Directly applying local and global Transformers will encounter two problems, *i.e.*, limited receptive fields and nontrivial computational costs. To address these challenges, we propose Half-Shuffle Transformer (HST) to play the role of $\mathcal{D}$.

**Network Architecture.** As shown in Fig. 3 (a), HST adopts a three-level U-shaped structure built by the basic unit Half-Shuffle Attention Block (HSAB). **Firstly**, HST uses a $conv3 \times 3$ to map reshaped $\mathbf{x}_k$ concatenated with stretched $\beta_k$ into feature $\mathbf{X}_0 \in \mathbb{R}^{H \times \hat{W} \times C}$, where $\hat{W} = W + d(N_\lambda - 1)$. **Secondly**, $\mathbf{X}_0$ passes through the encoder, bottleneck, and decoder to be embedded into deep feature $\mathbf{X}_d \in \mathbb{R}^{H \times \hat{W} \times C}$. Each level of the encoder or decoder contains an HSAB and a resizing module.

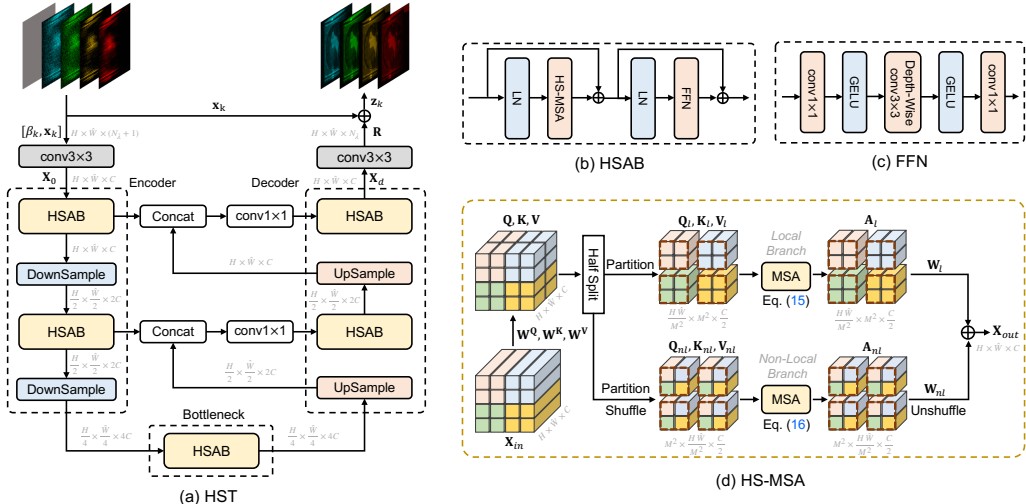

Figure 3: Diagram of HST. (a) HST adopts a U-shaped structure. (b) HSAB consists of an FFN, an HS-MSA, and two layer normalization. (c) Components of FFN. (d) HS-MSA contains *local branch* and *non-local branch*.

In Fig. 3 (b), HSAB consists of two layer normalization (LN), an HS-MSA, and a Feed-Forward Network (FFN) that is detailed in Fig. 3 (c). The downsampling and upsampling modules are strided $conv4 \times 4$ and $deconv2 \times 2$. **Finally**, a $conv3 \times 3$ operates on $\mathbf{X}_d$ to generate a residual image $\mathbf{R} \in \mathbb{R}^{H \times \hat{W} \times N_\lambda}$. The output denoised image $\mathbf{z}_k$ is obtained by the sum of $\mathbf{x}_k$ and reshaped $\mathbf{R}$.

**Half-Shuffle Multi-head Self-Attention.** The most important element of HSAB is the proposed Half-Shuffle Multi-head Self-Attention (HS-MSA) module. Fig. 3 (d) depicts the HS-MSA used in the first level. The input tokens of HS-MSA are denoted as $\mathbf{X}_{in} \in \mathbb{R}^{H \times \hat{W} \times C}$. Subsequently, $\mathbf{X}_{in}$ is linearly projected into $query$ $\mathbf{Q} \in \mathbb{R}^{H \times \hat{W} \times C}$, $key$ $\mathbf{K} \in \mathbb{R}^{H \times \hat{W} \times C}$, and $value$ $\mathbf{V} \in \mathbb{R}^{H \times \hat{W} \times C}$ as

$$\mathbf{Q} = \mathbf{X}_{in}\mathbf{W^Q}, \ \mathbf{K} = \mathbf{X}_{in}\mathbf{W^K}, \ \mathbf{V} = \mathbf{X}_{in}\mathbf{W^V}, \tag{13}$$

where $\mathbf{W^Q}, \mathbf{W^K}, \mathbf{W^V} \in \mathbb{R}^{C \times C}$ are learnable parameters and biases are omitted for simplification. Our HS-MSA combines the advantages of global MSA [42] and local window-based MSA [41], *i.e.*, HS-MSA can jointly capture local contextual information through the *local branch* and model long-range dependencies through the *non-local branch*, all while being computationally cheaper than global MSA. Specifically, $\mathbf{Q}, \mathbf{K}, \mathbf{V}$ are split into two equal parts along the channel dimension as

$$\mathbf{Q} = [\mathbf{Q}_l, \mathbf{Q}_{nl}], \ \mathbf{K} = [\mathbf{K}_l, \mathbf{K}_{nl}], \ \mathbf{V} = [\mathbf{V}_l, \mathbf{V}_{nl}], \tag{14}$$

where $\mathbf{Q}_l, \mathbf{K}_l, \mathbf{V}_l \in \mathbb{R}^{H \times \hat{W} \times \frac{C}{2}}$ are fed into the *local branch* to capture local contents, while $\mathbf{Q}_{nl}, \mathbf{K}_{nl}, \mathbf{V}_{nl} \in \mathbb{R}^{H \times \hat{W} \times \frac{C}{2}}$ pass through the *non-local branch* to model non-local dependencies.

**Local Branch.** The *local branch* computes MSA within position-specific windows. As shown in the upper path of Fig. 3 (d), $\mathbf{Q}_l, \mathbf{K}_l, \mathbf{V}_l$ are partitioned into non-overlapping windows of size $M \times M$. Then they are reshaped into $\mathbb{R}^{\frac{H\hat{W}}{M^2} \times M^2 \times \frac{C}{2}}$. Subsequently, $\mathbf{Q}_l, \mathbf{K}_l, \mathbf{V}_l$ are split along the channel wise into $h$ heads: $\mathbf{Q}_l = [\mathbf{Q}_l^1, \dots, \mathbf{Q}_l^h]$, $\mathbf{K}_l = [\mathbf{K}_l^1, \dots, \mathbf{K}_l^h]$, and $\mathbf{V}_l = [\mathbf{V}_l^1, \dots, \mathbf{V}_l^h]$. The dimension of each head is $d_h = \frac{C}{2h}$. Note that Fig. 3 (d) depicts the situation with $h = 1$ and some details are omitted for simplification. The local self-attention $\mathbf{A}_l^i$ is calculated inside each head as

$$\mathbf{A}_l^i = \text{softmax}\left(\frac{\mathbf{Q}_l^i \mathbf{K}_l^{i\mathsf{T}}}{\sqrt{d_h}} + \mathbf{P}_l^i\right)\mathbf{V}_l^i, \quad i = 1, \dots, h, \tag{15}$$

where $\mathbf{P}_l^i \in \mathbb{R}^{M^2 \times M^2}$ are learnable parameters embedding the position information.

**Non-local Branch.** The *non-local branch* computes cross-window interactions through shuffle operations inspired by ShuffleNet [59]. In particular, $\mathbf{Q}_{nl}, \mathbf{K}_{nl}, \mathbf{V}_{nl} \in \mathbb{R}^{H \times \hat{W} \times \frac{C}{2}}$ are firstly partitioned into non-overlapping windows with size $M \times M$. Then their shapes are transposed from $\mathbb{R}^{\frac{H\hat{W}}{M^2} \times M^2 \times \frac{C}{2}}$ to $\mathbb{R}^{M^2 \times \frac{H\hat{W}}{M^2} \times \frac{C}{2}}$ to shuffle the positions of tokens and establish inter-window dependencies. $\mathbf{Q}_{nl}, \mathbf{K}_{nl}, \mathbf{V}_{nl}$ are split into $h$ heads: $\mathbf{Q}_{nl} = [\mathbf{Q}_{nl}^1, \dots, \mathbf{Q}_{nl}^h]$, $\mathbf{K}_{nl} = [\mathbf{K}_{nl}^1, \dots, \mathbf{K}_{nl}^h]$,

| Algorithms | Params | GFLOPS | S1 | S2 | S3 | S4 | S5 | S6 | S7 | S8 | S9 | S10 | Avg |
|---|---|---|---|---|---|---|---|---|---|---|---|---|---|
| TwIST [60] | - | - | 25.16
0.700 | 23.02
0.604 | 21.40
0.711 | 30.19
0.851 | 21.41
0.635 | 20.95
0.644 | 22.20
0.643 | 21.82
0.650 | 22.42
0.690 | 22.67
0.569 | 23.12
0.669 |
| GAP-TV [26] | - | - | 26.82
0.754 | 22.89
0.610 | 26.31
0.802 | 30.65
0.852 | 23.64
0.703 | 21.85
0.663 | 23.76
0.688 | 21.98
0.655 | 22.63
0.682 | 23.10
0.584 | 24.36
0.669 |
| DeSCI [23] | - | - | 27.13
0.748 | 23.04
0.620 | 26.62
0.818 | 34.96
0.897 | 23.94
0.706 | 22.38
0.683 | 24.45
0.743 | 22.03
0.673 | 24.56
0.732 | 23.59
0.587 | 25.27
0.721 |
| $\lambda$-Net [33] | 62.64M | 117.98 | 30.10
0.849 | 28.49
0.805 | 27.73
0.870 | 37.01
0.934 | 26.19
0.817 | 28.64
0.853 | 26.47
0.806 | 26.09
0.831 | 27.50
0.826 | 27.13
0.816 | 28.53
0.841 |
| HSSP [35] | - | - | 31.48
0.858 | 31.09
0.842 | 28.96
0.823 | 34.56
0.902 | 28.53
0.808 | 30.83
0.877 | 28.71
0.824 | 30.09
0.881 | 30.43
0.868 | 28.78
0.842 | 30.35
0.852 |
| DNU [34] | **1.19M** | 163.48 | 31.72
0.863 | 31.13
0.846 | 29.99
0.845 | 35.34
0.908 | 29.03
0.833 | 30.87
0.887 | 28.99
0.839 | 30.13
0.885 | 31.03
0.876 | 29.14
0.849 | 30.74
0.863 |
| DIP-HSI [30] | 33.85M | 64.42 | 32.68
0.890 | 27.26
0.833 | 31.30
0.914 | 40.54
0.962 | 29.79
0.900 | 30.39
0.877 | 28.18
0.913 | 29.44
0.874 | 34.51
0.927 | 28.51
0.851 | 31.26
0.894 |
| TSA-Net [20] | 44.25M | 110.06 | 32.03
0.892 | 31.00
0.858 | 32.25
0.915 | 39.19
0.953 | 29.39
0.884 | 31.44
0.908 | 30.32
0.878 | 29.35
0.888 | 30.01
0.890 | 29.59
0.874 | 31.46
0.894 |
| DGSMP [38] | 3.76M | 646.65 | 33.26
0.915 | 32.09
0.898 | 33.06
0.925 | 40.54
0.964 | 28.86
0.882 | 33.08
0.937 | 30.74
0.886 | 31.55
0.923 | 31.66
0.911 | 31.44
0.925 | 32.63
0.917 |
| GAP-Net [36] | 4.27M | 78.58 | 33.74
0.911 | 33.26
0.900 | 34.28
0.929 | 41.03
0.967 | 31.44
0.919 | 32.40
0.925 | 32.27
0.902 | 30.46
0.905 | 33.51
0.915 | 30.24
0.895 | 33.26
0.917 |
| ADMM-Net [37] | 4.27M | 78.58 | 34.12
0.918 | 33.62
0.902 | 35.04
0.931 | 41.15
0.966 | 31.82
0.922 | 32.54
0.924 | 32.42
0.896 | 30.74
0.907 | 33.75
0.915 | 30.68
0.895 | 33.58
0.918 |
| HDNet [32] | 2.37M | 154.76 | 35.14
0.935 | 35.67
0.940 | 36.03
0.943 | 42.30
0.969 | 32.69
0.946 | 34.46
0.952 | 33.67
0.926 | 32.48
0.941 | 34.89
0.942 | 32.38
0.937 | 34.97
0.943 |
| MST-L [61] | 2.03M | 28.15 | 35.40
0.941 | 35.87
0.944 | 36.51
0.953 | 42.27
0.973 | 32.77
0.947 | 34.80
0.955 | 33.66
0.925 | 32.67
0.948 | 35.39
0.949 | 32.50
0.941 | 35.18
0.948 |
| MST++ [62] | 1.33M | 19.42 | 35.80
0.943 | 36.23
0.947 | 37.34
0.957 | 42.63
0.973 | 33.38
0.952 | 35.38
0.957 | 34.35
0.934 | 33.71
0.953 | 36.67
0.953 | 33.38
0.945 | 35.99
0.951 |
| CST-L [62] | 3.00M | 40.01 | 35.96
0.949 | 36.84
0.955 | 38.16
0.962 | 42.44
0.975 | 33.25
0.955 | 35.72
0.963 | 34.86
0.944 | 34.34
0.961 | 36.51
0.957 | 33.09
0.945 | 36.12
0.957 |
| BIRNAT [63] | 4.40M | 2122.66 | 36.79
0.951 | 37.89
0.957 | 40.61
0.971 | **46.94**
**0.985** | 35.42
0.964 | 35.30
0.959 | 36.58
0.955 | 33.96
0.956 | 39.47
0.970 | 32.80
0.938 | 37.58
0.960 |
| **DAUHST-2stg** | 1.40M | **18.44** | 35.93
0.943 | 36.70
0.946 | 37.96
0.959 | 44.38
0.978 | 34.13
0.954 | 35.43
0.957 | 34.78
0.940 | 33.65
0.950 | 37.42
0.955 | 33.07
0.941 | 36.34
0.952 |
| **DAUHST-3stg** | 2.08M | 27.17 | 36.59
0.949 | 37.93
0.958 | 39.32
0.964 | 44.77
0.980 | 34.82
0.961 | 36.19
0.963 | 36.02
0.950 | 34.28
0.956 | 38.54
0.963 | 33.67
0.947 | 37.21
0.959 |
| **DAUHST-5stg** | 3.44M | 44.61 | 36.92
0.955 | 38.52
0.962 | 40.51
0.967 | 45.09
0.980 | 35.33
0.964 | 36.56
0.965 | 36.82
0.958 | 34.74
0.959 | 38.71
0.963 | 34.27
0.952 | 37.75
0.962 |
| **DAUHST-9stg** | 6.15M | 79.50 | **37.25**
**0.958** | **39.02**
**0.967** | **41.05**
**0.971** | 46.15
0.983 | **35.80**
**0.969** | **37.08**
**0.970** | **37.57**
**0.963** | **35.10**
**0.966** | **40.02**
**0.970** | **34.59**
**0.956** | **38.36**
**0.967** |

Table 1: Comparisons between DAUHST and SOTA methods on 10 simulation scenes (S1~S10). Params, FLOPS, PSNR (upper entry in each cell), and SSIM (lower entry in each cell) are reported.

and $\mathbf{V}_{nl} = [\, \mathbf{V}_{nl}^1, \dots, \mathbf{V}_{nl}^h \,]$. Then the non-local self-attention $\mathbf{A}_{nl}^i$ is computed in each head as

$$\mathbf{A}_{nl}^i = \mathrm{softmax}\big(\frac{\mathbf{Q}_{nl}^i \, {\mathbf{K}_{nl}^i}^{\mathsf{T}}}{\sqrt{d_h}} + \mathbf{P}_{nl}^i\big) \mathbf{V}_{nl}^i, \quad i = 1, \dots, h, \tag{16}$$

where $\mathbf{P}_{nl}^i \in \mathbb{R}^{\frac{H\hat{W}}{M^2} \times \frac{H\hat{W}}{M^2}}$ are learnable parameters representing the position embedding. Subsequently, $\mathbf{A}_{nl}^i \in \mathbb{R}^{M^2 \times \frac{H\hat{W}}{M^2} \times d_h}$ is unshuffled by being transposed to shape $\mathbb{R}^{\frac{H\hat{W}}{M^2} \times M^2 \times d_h}$. Then the outputs of *local branch* in Eq. (15) and *non-local branch* in Eq. (16) are aggregated by a linear projection as

$$\mathrm{HS\text{-}MSA}(\mathbf{X}_{in}) = \sum_{i=1}^h \mathbf{A}_l^i \mathbf{W}_l^i + \sum_{i=1}^h \mathbf{A}_{nl}^i \mathbf{W}_{nl}^i, \tag{17}$$

where $\mathbf{W}_l^i, \mathbf{W}_{nl}^i \in \mathbb{R}^{d_h \times C}$ refer to learnable parameters. We reshape the result of Eq. (17) to obtain the output $\mathbf{X}_{out} \in \mathbb{R}^{H \times \hat{W} \times C}$. Instead of globally sampling all tokens, HS-MSA builds inter-window correlations by shuffle operations. The self-attention is calculated in the local window but with tokens from non-local regions. Therefore, HS-MSA is much computationally cheaper than global MSA.

## 3 Experiment

### 3.1 Experiment Setup

Similar to [20, 32, 36, 38, 61], 28 wavelengths are selected from 450nm to 650nm and derived by spectral interpolation manipulation for the HSI data. Simulation and real experiments are conducted.

**Simulation Dataset.** We adopt two datasets, *i.e.*, CAVE [64] and KAIST [65] for simulation experiments. The CAVE dataset consists of 32 HSIs with spatial size $512 \times 512$. The KAIST dataset contains 30 HSIs of spatial size $2704 \times 3376$. Following the settings of [20, 32, 36, 38, 61], the CAVE dataset is adopted as the training set while 10 scenes from the KAIST dataset are selected for testing.

**Real Dataset.** Five real HSIs collected by the CASSI system developed in [20] are used for testing.

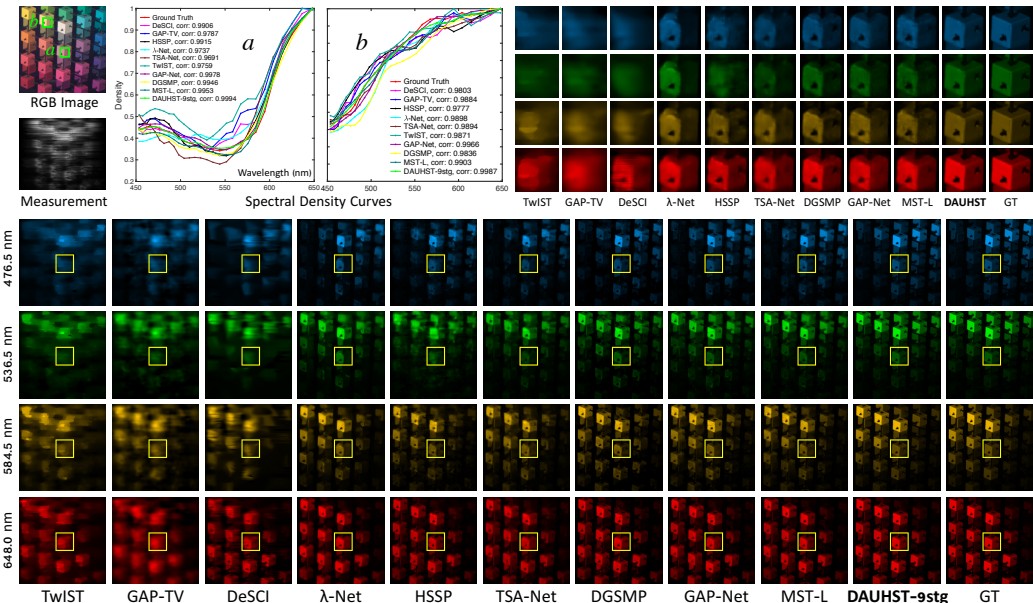

Figure 4: Simulation HSI reconstruction comparisons of *Scene* 2 with 4 (out of 28) spectral channels. The top-middle shows the spectral curves corresponding to the two green boxes of the RGB image. The top-right depicts the enlarged patches corresponding to the yellow boxes in the bottom HSIs. Zoom in for a better view.

**Implementation Details.** We implement DAUHST by Pytorch. All DAUHST models are trained with Adam [66] optimizer ($\beta_1 = 0.9$ and $\beta_2 = 0.999$) using Cosine Annealing scheme [67] for 300 epochs on an RTX 3090 GPU. The initial learning rate is $4 \times 10^{-4}$. Patches with spatial sizes $256 \times 256$ and $660 \times 660$ are randomly cropped from the 3D HSI cubes with 28 channels as training samples for the simulation and real experiments. The shifting step $d$ in the dispersion is set to 2. The batch size is 5. We set the basic channel $C = N_\lambda = 28$ to store HSI information. The weights of $\mathcal{D}$ in different stages are unshared. Data augmentation includes random rotation and flipping. The training objective is to minimize the Root Mean Square Error (RMSE) between reconstructed and ground-truth HSIs.

## 3.2 Quantitative Comparisons with State-of-the-Art Methods

Tab. 1 compares the results of DAUHST and 16 SOTA methods including three model-based methods (TwIST [60], GAP-TV [26], and DeSCI [23]), one PnP method (DIP-HSI [30]), seven E2E methods ($\lambda$-Net [33], TSA-Net [20], HDNet [32], MST [61], MST++ [62], CST [68], and BIRNAT [63]), and five deep unfolding methods (HSSP [35], DNU [34], DGSMP [38], GAP-Net [36], and ADMM-Net [37]) on 10 simulation scenes. All algorithms are tested with the same settings as [38, 61].

**(i)** Our best model DAUHST-9stg (9-stage DAUHST) yields very impressive results, *i.e.*, 38.36 dB in PSNR and 0.967 in SSIM. DAUHST-9stg significantly outperforms two recent SOTA methods BIRNAT [63] and MST-L [61] by 0.78 and 3.18 dB, suggesting the effectiveness of our method.

**(ii)** Our DAUHST models dramatically surpass SOTA methods while requiring cheaper computational and memory costs. For instance, when compared with the only one Transformer-based E2E method MST, our DAUHST-2stg outperforms MST-L by 1.16 dB but only costs 68.9% (1.40 / 2.03) Params and 65.5% (18.44 / 28.15) FLOPS. When compared with CNN-based E2E methods, DAUHST-3stg surpasses HDNet, TSA-Net, and $\lambda$-Net by 2.24, 5.75, and 8.68 dB while only requiring 87.8%, 4.7%, 3.3% Params and 17.6%, 24.7%, 23.0% FLOPS. When compared with RNN-based E2E method BIRNAT, our DAUHST-5stg is 0.17 dB higher but only costs 2.1% FLOPS and 78.2% Params. Fig. 1 plots the PSNR-FLOPS comparisons of DAUHST and SOTA unfolding methods. DAUHST outperforms other competitors with the same number of stages by very large margins, **over 4 dB**.

## 3.3 Qualitative Comparisons with State-of-the-Art Methods

**Simulation HSI Reconstruction.** Fig. 4 depicts the simulation HSI reconstruction comparisons between our DAUHST and other SOTA methods on *Scene* 2 with 4 (out of 28) spectral channels. The top-right part shows the zoomed-in patches of the yellow boxes in the entire HSIs (bottom). As can be observed that our DAUHST-9stg is more favorable to reconstruct visually pleasant HSIs with more

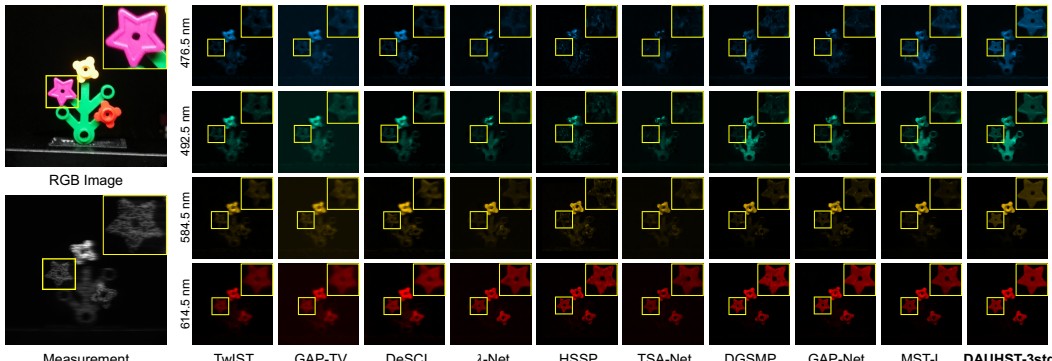

Figure 5: Real HSI reconstruction results of DAUHST-3stg and 9 SOTA methods on *Scene* 1 with 4 (out of 28) spectra. Only our method can clearly reconstruct the picked flower at all wavelengths. Zoom in for a better view.

detailed contents, cleaner textures, and fewer artifacts while preserving the spatial smoothness of homogeneous regions. In contrast, previous methods either yield over-smooth results compromising fine-grained structures, or introduce undesired chromatic artifacts and blotchy textures that are absent in the ground truth (GT). The top-middle part illustrates the density-wavelength spectral curves corresponding to the green boxes identified as *a* and *b* in the RGB image (top-left). The spectral curves of DAUHST-9stg achieve the highest correlation and coincidence with the reference curves, showing the advantage of our proposed DAUHST in spectral-dimension consistency reconstruction.

**Real HSI Reconstruction.** We further evaluate the effectiveness of DAUHST in real HSI reconstruction. Following the same settings as [20, 38, 61] for a fair comparison, we re-train DAUHST-3stg with the real mask on the CAVE and KAIST datasets jointly. To simulate the real imaging situations, the training samples are also injected with 11-bit shot noise. Fig. 5 shows the visual comparisons between our DAUHST-3stg and nine SOTA methods. In the top three rows, only our DAUHST-3stg can reconstruct the flower patch corresponding to the yellow box at all wavelengths while other methods all fail to recover the entire patch. In the bottom row, DAUHST-3stg restores more HSI structural details and clearer contents with fewer artifacts. In contrast, other methods recover blurry images, generate incomplete responses, and are susceptible to the noise corruption. This evidence suggests that DAUHST is more robust to the noise distortion and more effective in real HSI reconstruction.

## 3.4 Ablation Study

**Break-down Ablation.** We adopt baseline-1 that is derived by removing HS-MSA and DAUF from DAUHST-3stg to conduct the break-down ablation. Our goal is to study the effect of each component towards higher performance. Baseline-1 is cascaded end to end by three single-stage networks. As shown in Tab. 2a, baseline-1 achieves 33.05 dB. When we respectively apply DAUF and HS-MSA, the model achieves 2.32 and 2.44 dB improvements. When we exploit DAUF and HS-MSA jointly, the model gains by 4.16 dB. These results demonstrate the effectiveness of our DAUF and HS-MSA.

**Self-Attention Mechanism.** To compare HS-MSA with other MSAs, we adopt baseline-2 that is obtained by removing HS-MSA from DAUHST-1stg to conduct the ablation in Tab. 2b. We remove different position embedding schemes to avoid their impacts and only compare MSAs. For fairness, we keep the Params of MSAs the same by fixing the number of channels and heads. Baseline-2 yields 32.79 dB. We apply global MSA (G-MSA) [42], Swin MSA (SW-MSA) [41], Spectral-wise MSA (S-MSA) [61], and HS-MSA. Note that we downsample the input feature maps of G-MSA to avoid memory bottlenecks. As shown in Tab. 2b, HS-MSA yields the most significant improvement of 1.26 dB, which is 0.42, 0.30, and 0.23 dB higher than G-MSA, SW-MSA, and S-MSA. This superiority is mainly derived from HS-MSA's ability to jointly capture local contents and non-local dependencies.

**Unfolding Framework.** We compare our DAUF with previous unfolding frameworks including DNU [34], ADMM-Net [37], and GAP-Net [36]. For a fair comparison, we replace each single-stage network of DNU, ADMM-Net, and GAP-Net by our HST. 3-stage architecture is adopted to conduct ablations. The results are shown in Tab. 2c. Our DAUF significantly outperforms DNU, ADMM, and GAP by 2.59, 1.69, and 1.63 dB while adding only 0.05M Params and 0.94G FLOPS. This is mainly because DAUF uses the parameters estimated from the compressed measurement and physical mask in the CASSI system to direct the iterative learning. These parameters capture critical information of CASSI degradation patterns and ill-posedness degree, providing key cues for HSI reconstruction.

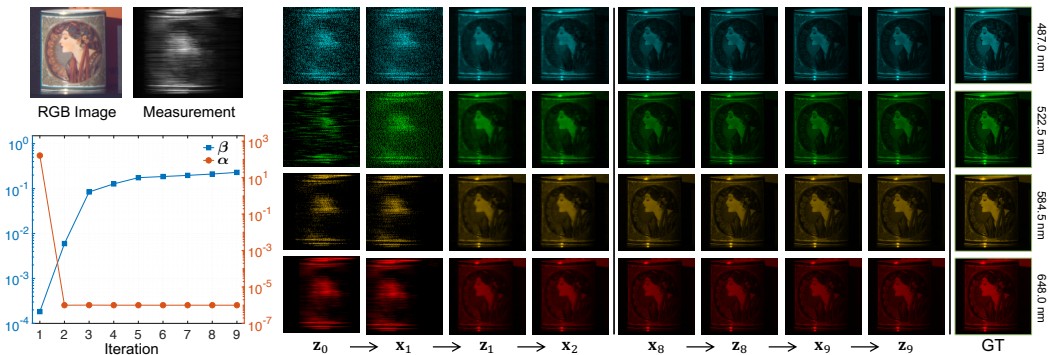

Figure 6: Visualization of $\mathbf{z}_k$ and $\mathbf{x}_k$ with 4 (out of 28) spectral channels on $Scene$ 1 in different iterations. The bottom-left corner plots the curves of $\boldsymbol{\alpha}$ and $\boldsymbol{\beta}$ changing with the iteration. Please zoom in for a better view.

| Baseline-1 | DAUF | HS-MSA | PSNR | SSIM | Params (M) | FLOPS (G) |
|---|---|---|---|---|---|---|
| ✓ | | | 33.05 | 0.912 | 1.06 | 17.62 |
| ✓ | ✓ | | 35.37 | 0.938 | 1.11 | 18.55 |
| ✓ | | ✓ | 35.49 | 0.941 | 2.03 | 26.23 |
| ✓ | ✓ | ✓ | **37.21** | **0.959** | 2.08 | 27.17 |

(a) Break-down ablation towards higher performance.

| Method | Baseline-2 | G-MSA | SW-MSA | S-MSA | **HS-MSA** |
|---|---|---|---|---|---|
| PSNR | 32.79 | 33.63 | 33.75 | 33.82 | **34.05** |
| SSIM | 0.904 | 0.920 | 0.924 | 0.926 | **0.930** |
| Params (M) | 0.40 | 0.48 | 0.48 | 0.48 | 0.48 |
| FLOPS (G) | 6.85 | 10.30 | 9.41 | 8.89 | 9.72 |

(b) Ablation of various self-attention mechanisms.

| Framework | DNU [34] | ADMM [37] | GAP [36] | **DAUF** |
|---|---|---|---|---|
| PSNR | 34.62 | 35.52 | 35.58 | **37.21** |
| SSIM | 0.930 | 0.942 | 0.943 | **0.959** |
| Params (M) | 2.03 | 2.03 | 2.03 | 2.08 |
| FLOPS (G) | 26.23 | 26.23 | 26.23 | 27.17 |

(c) Ablation of different unfolding frameworks.

| Baseline-3 | $\alpha$ | $\beta$ | PSNR | SSIM | Params (M) | FLOPS (G) |
|---|---|---|---|---|---|---|
| ✓ | | | 36.49 | 0.952 | 2.03 | 26.23 |
| ✓ | ✓ | | 36.94 | 0.957 | 2.08 | 27.10 |
| ✓ | | ✓ | 36.83 | 0.956 | 2.08 | 27.17 |
| ✓ | ✓ | ✓ | **37.21** | **0.959** | 2.08 | 27.17 |

(d) Ablation to study the effect of parameters $\boldsymbol{\alpha}$ and $\boldsymbol{\beta}$.

Table 2: Ablation studies on simulation datasets [64, 65]. PSNR, SSIM, Params, and FLOPS are reported.

To study the effect of the estimated parameters $\boldsymbol{\alpha}$ and $\boldsymbol{\beta}$, we perform a break-down ablation of DAUF. We adopt DAUHST-3stg as baseline-3 but $\boldsymbol{\alpha}$ is set as learnable parameters instead of being estimated by $\mathcal{E}$ in Eq. (12) and $\boldsymbol{\beta}$ is not fed into $\mathcal{D}$. The results are shown in Tab. 2d. Baseline-3 yields 36.49 dB. When $\boldsymbol{\alpha}$ is set to be estimated by $\mathcal{E}$, baseline-3 is improved by 0.45 dB. When $\boldsymbol{\beta}$ is fed into $\mathcal{D}$, baseline-3 gains by 0.34 dB. When $\boldsymbol{\alpha}$ and $\boldsymbol{\beta}$ are exploited jointly in the iterative learning, baseline-3 achieves a significant improvement of 0.72 dB. These results verify that the estimated parameters $\boldsymbol{\alpha}$ and $\boldsymbol{\beta}$ are beneficial for the linear projection and denoising network of deep unfolding methods.

To further analyze the roles of the estimated parameters, we visualize $\mathbf{z}_k$ and $\mathbf{x}_k$ of Eq. (12), and plot the curves of $\boldsymbol{\alpha}$ and $\boldsymbol{\beta}$ as they change with the iteration in Fig. 6. We observe: **(i)** $\mathbf{z}_0$ and $\mathbf{x}_1$ yield either blurry or noisy images. There is a significant gap between them. Since $\alpha_k = \mu_k$ in Eq. (5) penalizes the differences between $\mathbf{z}$ and $\mathbf{x}$, $\alpha_1$ is estimated to be a large value. From the linear projection of the second iteration ($\mathbf{z}_1 \to \mathbf{x}_2$) on, the gap between $\mathbf{z}$ and $\mathbf{x}$ decreases substantially. Therefore, $\alpha_k$ are estimated to be small values when $k \geq 2$. This indicates that $\boldsymbol{\alpha}$ can adaptively scale the linear projection $\mathcal{P}$. **(ii)** The noise corruption is severe in the first iteration. Thus, $\beta_1 = \mu_1/\tau_1 = 1/(\sqrt{\tau_1/\mu_1})^2$, which is inversely proportional to the noise level, is estimated to be a small value. With further iterations, the noise level decreases, and thus the estimated $\beta_k$ increases. These results demonstrate that $\boldsymbol{\beta}$ can provide the information about noise level for the denoising network $\mathcal{D}$.

## 4 Conclusion

In this paper, we remedy two issues of previous deep unfolding methods, *i.e.*, they do not estimate informative parameters from the CASSI system to direct the iterative learning and they are mainly CNN-based showing limitations in capturing long-range dependencies. To cope with these challenges, we firstly formulate a principled MAP-based unfolding framework DAUF that estimates parameters from the compressed measurement and physical mask. Then the parameters, which capture critical cues of CASSI degradation patterns and ill-posedness degree, are fed into each iteration to contextually scale the linear projection and provide noise level information for the denoising network. Secondly, we propose a novel Transformer HST that can jointly extract local contents and model non-local dependencies. By plugging HST into DAUF, we derive the first Transformer-based unfolding method DAUHST for HSI reconstruction. Comprehensive experiments show that our DAUHST outperforms SOTA methods by a large margin while requiring much cheaper memory and computational costs.

## Limitation and Social Impact

The main limitation of our work is that the performance improvement of our method comes with lowering down the inference speed and increasing the model complexity. Until now, spectral snapshot compressive imaging reconstruction techniques have no negative social impact yet. Our proposed DAUHST does not present any negative foreseeable societal consequence, either.

## Acknowledgement

This work is partially supported by the NSFC fund (61831014), the Shenzhen Science and Technology Project under Grant (JSGG20210802153150005, CJGJZD20200617102601004). Xin Yuan acknowledges the support of NSFC (62271414), Westlake Foundation (2021B1501-2) and the Research Center for Industries of the Future (RCIF) at Westlake University.

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
