# OpenReview forum: "Degradation-Aware Unfolding Half-Shuffle Transformer for Spectral Compressive Imaging"
_NeurIPS.cc/2022/Conference — NeurIPS 2022 Accept_

### Official Review · Reviewer_6Fb8 · 2022-07-11

**Rating:** 5
**Confidence:** 3
**Soundness:** 2 fair
**Presentation:** 3 good
**Contribution:** 2 fair

**Summary:**

this submission deals with designing unfolding based architecture for supervised learning of spectral compressive imaging (SCI). Compared with SOTA, it proposes to explicitly integrate the ill-posedness of the acquisition (degradation and mask) into each block (iteration) for data consistency and denoising. The denoising network also uses a U-shape transformer with half-shuffle multihead self-attention to capture both local and non-local contents. The experiments outperform SOTA in terms of both memory, computer, and reconstruction fidelity.

**Questions:**

This work can be strengthened with more ablations that explains the design choices for the U-shaped transformer. For example, what if one simply uses a multi-head vision transformer instead of U-shaped architecture?


**Limitations:**


The limitations are not included in the submission. The limitations need to be discussed. For example, it seems that the entire network needs to be trained from scratch each time the measurement model changes. The robustness of the model also needs to be discussed in terms of introducing hallucinations.

**Strengths And Weaknesses:**

Strength
practical significance: the gains achieved are significant compared with the reported SOTA methods
practical novelty: this is the first work deploying transformers as the denoiser for SCI.
clear writing: the paper is well written and the idea is clear

Weakness
limited technical novelty: The main novelty of this work is to propose a U-shaped transformer with half–shuffle multi-head self-attention for the denoising network in deep unfolding architectures for SCI. Deep unfolding is well established. Also, there are several works reporting superiority of transformers against CNNs for denoising and low level computer vision tasks, and it is not surprising to observe similar gains for SCI; see e.g., [Peit et al’21].

[Peit et al’21]Petit O, Thome N, Rambour C, Themyr L, Collins T, Soler L. U-net transformer: Self and cross attention for medical image segmentation. InInternational Workshop on Machine Learning in Medical Imaging 2021 Sep 27 (pp. 267-276). Springer, Cham.

---

> ### Author Response · Authors · 2022-07-30
> **Response to Reviewer 6Fb8 - Part 3**
>
> `Q-4:` Question about discussing the limitations of this work
>
> `A-4:` In fact, we have discussed the limitations of our work in Sec. 5 of the supplementary. The main limitation of our work is that the performance improvement of our method comes with lowering down the inference speed and increasing the model complexity. Specifically, the Params, FLOPS, and depth of network increase with the stage number of DAUHST. For instance, compared with DAUHST-1stg, DAUHST-9stg achieves 4.00 dB improvement but requires 8.18$\times$ FLOPS, 8.42$\times$ Params, and 4.04$\times$ inference time. To tackle this limitation, we will study how to improve the restoration performance without increasing the model complexity and sacrificing the inference speed. However, as shown in Tab. 1 and Sec. 3.2 of the main paper, our DAUHST significantly outperforms other state-of-the-art methods while using cheaper computational and memory costs.
>
> `Q-5:` Question about evaluating the robustness of the model
>
> `A-5:` In fact, we have done ablation to evaluate the robustness of our method in Sec. 4.2 and Tab. 1(b) of the supplementary. After training the network, we fix its parameters. In the testing phase, we change the mask by randomly cropping it with size 256×256 from the real mask of size 660×660 to evaluate the robustness of DAUHST for different signal modulations. We directly copy the results in Tab. 1(b) of the supplementary to the following table, where Mask-0 indicates the original mask used in the training phase. Compared with the two SOTA methods TSA-Net that degrades by 7.51% on average and DGSMP that degrades by 13.91% on average, our DAUHST-3stg declines by much smaller margins, i.e., only 2.03% on average. These results suggest that DAUHST does not need to be re-trained from scratch when the measurement model changes because DAUHST only sacrifices marginal performance that can be ignored. In other words, our DAUHST is more robust and flexible for large-scale SCI reconstruction.
>
> | Method| TSA-Net| DGSMP | DAUHST-3stg |
> | - | - | - | - |
> | Mask-0 | 31.46 $\downarrow$ `0.00%` | 32.63 $\downarrow$ `0.00%` | 37.21 $\downarrow$ `0.00%` |
> | Mask-1 | 29.18 $\downarrow$ `7.24%` | 28.50 $\downarrow$ `12.66%` | 36.43 $\downarrow$ `2.10%`|
> | Mask-2 | 29.10 $\downarrow$ `7.50%` | 27.87 $\downarrow$ `14.59%` | 36.55 $\downarrow$ `1.77%`|
> | Mask-3 | 29.01 $\downarrow$ `7.79%` | 27.91 $\downarrow$ `14.47%` | 36.38 $\downarrow$ `2.23%`|

---

> > ### Comment · Reviewer_6Fb8 · 2022-08-07
> > **some concerns addressed, but still believe this work lacks technical novelty**
> >
> > I would like to thank the authors for their effort to provide an extensive response in explaining the technical novelties and providing new experiments. It is clarified now that the proposed U-shaped architecture is superior to alternative U-shaped transformers and ViT for denoising  and the design choices are important based on the provided ablation. Thus, I raise my score by one.
> >
> > Although this work has practical significance, I still believe that this work has limited technical novelty. This work relies on deep unfolding/unrolled networks for inverse imaging, and guiding the unfolding with the ill-posedness of the measurement operator is not significant.

---

> > > ### Author Response · Authors · 2022-08-09
> > > **Discussions with Reviewer 6Fb8**
> > >
> > > Thanks for your reply. As you mention that “guiding the unfolding
> > > with the ill-posedness of the measurement operator is not significant.” We would like to explain the contribution of our unfolding framework DAUF to you.
> > >
> > > As we analyze in `A-1` (a) of ` Response to Reviewer 6Fb8 - Part 1`, Our DAUF is fundamentally different from existing deep unfolding frameworks. Previous unfolding methods do not estimate the degradation patterns and ill-posedness degree from the highly related CASSI system to guide the iterative learning. In contrast, we formulate a principled degradation-aware unfolding framework (DAUF) to address this issue. our DAUF estimates parameters from the compressed measurement and physical mask in the CASSI system. Then the parameters, which capture critical cues of CASSI degradation patterns and ill-posedness degree, are fed into each iteration to contextually scale the linear projection and provide noise level information for the denoising network.
> > >
> > > $\textbf{Our degradation guiding scheme does significantly improve the performance.}$ In Table 2(c) of our main paper, we conduct ablation study to compare our DAUF with different unfolding frameworks. We also list the results here in the following table. As can be seen that our DAUF dramatically surpasses previous deep unfolding frameworks DNU, ADMM, and GAP by 2.59, 1.69, and 1.63 dB. This evidence demonstrates the impressive effectiveness and contribution of our DAUF for deep unfolding framework.
> > >
> > > | Framework | DNU | ADMM | GAP | DAUF |
> > > | - | - | - | - | - |
> > > | PSNR | 34.62 | 35.52 | 35.58 | 37.21 |
> > > | SSIM | 0.930 | 0.942 | 0.943 | 0.959 |
> > >
> > > We hope to further discuss with you whether your concerns have been addressed or not. If you still have any unclear parts of our work, please let us know. Thanks.
> > >
> > > Best regards,
> > >
> > > Authors

---

> > ### Author Response · Authors · 2022-08-09
> > **Thank Reviewer 6Fb8 for approving our work**
> >
> > Dear Reviewer 6Fb8,
> >
> > Thanks for discussing with us and agreeing that our response addresses some of your concerns. Thank you for approving the $\textbf{contribution}$ of our proposed Transformer and $\textbf{practical significance}$ of our work. We will kindly cite and introduce U-Transformer in the revision.
> >
> > Best regards,
> >
> > Authors

---

> ### Author Response · Authors · 2022-07-30
> **Response to Reviewer 6Fb8 - Part 2**
>
> `Q-2:` Comparison with U-Transformer [1]
>
> `A-2:` We argue that our DAUHST is completely different from U-Transformer. In fact, the only thing they have in common is that they both adopt U-shaped structure. Yet, the U-shaped structure is proposed by U-Net [2] and has been widely applied. We do not claim this as our contribution. Our DAUHST and U-Transformer are different in
>
> (a) Motivation and research topic
>
> U-Transformer aims to address the conceptual limitations and model spatial relationships between anatomical structures. Our DAUHST is proposed to perceive CASSI degradation patterns to adjust the iterative learning, and jointly capture local and non-local dependencies of spatially sparse HSI representations. U-Transformer is designed for medical image segmentation while DAUHST is customized for SCI reconstruction
>
> (b) Technique
>
> (i) We firstly focus on comparing the self-attention mechanism. U-Transformer mainly proposes two attentions, multi-head self-attention (MHSA) and multi-head cross-attention (MHCA). MHCA is applied to the skip connection between encoder and decoder to filter out non-semantic features and allow a fine spatial recovery. It is not self-attention. The MHSA is a vanilla global multi-head self-attention. Its computational costs are quadratic to the spatial dimension of the input map. This is a huge burden and sometimes unaffordable. Thus, only a single MHSA is applied at the lowest resolution of the encoder in U-Transformer. In contrast, as analyzed in Sec. 2.3, our HS-MSA jointly develops two branches. The local branch captures local contents by computing self-attention within local patches. The non-local branch models long-range dependencies by globally shuffling tokens and then calculating self-attention between tokens from different patches. Thus, the computational complexity of our HS-MSA is much lower than MHSA. Besides, U-Transformer only employs a single MHSA layer. In contrast, DAUHST plugs HS-MSA into the basic block as a key component to build up the whole network. So the long-range dependencies can be repeatedly captured and enhanced by DAUHST. Hence, our DAUHST can model more robust and stronger non-local interactions than U-Transformer
>
> (ii) All details of DAUHST and U-Transformer are completely different. For instance, the components of downsampling and upsampling. U-Transformer has four levels while DAUHST has three. DAUHST is unfolded from the MAP energy function based on SCI degradation model in Eq. (1) and (2) of the paper. While U-Transformer learns an end-to-end implicit mapping of segmentation. Thus, DAUHST is more intuitively interpretable by explicitly characterizing the image priors and the CASSI system imaging model. DAUHST uses layer normalization while U-Transformer uses batch normalization. And so on. In a nutshell, DAUHST and U-Transformer are two completely different things
>
> (c) Experiment
>
> We conduct experiments to compare DAUHST and U-Transformer in the following table. Our DAUHST-3stg dramatically surpasses U-Transformer by 6.14 dB while only costing 21% FLOPS, suggesting the significant superiority of DAUHST over U-Transformer.
>
> | Method| PSNR | SSIM | Params (M) | FLOPS (G) |
> | - | - | - | - | - |
> | U-Transformer | 31.07 | 0.861 | 6.79 | 129.21 |
> | DAUHST-3stg | 37.21 | 0.959 | 2.08 | 27.17 |
>
> We will cite [1] and add these results in the revision
>
> `Q-3:` Ablations of the design choices for the U-shaped structure
>
> `A-3:` Thanks for your advice, we conduct the ablation as you recommend. Specifically, we simply apply a multi-head vision Transformer, i.e., ViT [3]. Besides, we also change the overall architecture to a single-scale structure without downsampling and upsampling. Three-stage structure is adopted. The results are listed in the following table, where $\dagger$ denotes using single-scale architecture, OOM indicates out of memory issue raised by GPU, and ViT* denotes downsampling the input images to $\frac{1}{4}$ size due to the memory limitation. As can be seen that the computational cost of directly applying ViT is unaffordable for our GPU. DAUHST$^\dagger$ yields 1.44 dB improvement than ViT*$^\dagger$ while costing less FLOPS, suggesting the effectiveness of HS-MSA. DAUHST surpasses DAUHST$^\dagger$ by 0.92 dB. This evidence shows that U-shape is a better choice of architecture for SCI reconstruction
>
> | Method| PSNR | SSIM | Params (M) | FLOPS (G) |
> | - | - | - | - | - |
> | ViT$^\dagger$ | OOM | OOM | OOM | OOM |
> | ViT*$^\dagger$ | 34.85 | 0.933 | 0.35 | 36.12 |
> | DAUHST$^\dagger$ | 36.29 | 0.951 | 0.35 | 27.30 |
> | DAUHST | 37.21 | 0.959 | 2.08 | 27.17 |
>
> We will add this ablation in the revision
>
> `Reference`
>
> [1] U-net transformer: Self and cross attention for medical image segmentation. In International Workshop on Machine Learning in Medical Imaging 2021.
>
> [2] U-net: Convolutional networks for biomedical image segmentation. In MICCAI 2015.
>
> [3] An Image is Worth 16x16 Words: Transformers for Image Recognition at Scale. In ICLR 2021.

---

> ### Author Response · Authors · 2022-07-30
> **Response to Reviewer 6Fb8 - Part 1**
>
> Thanks for your valuable comments, now we answer your questions one by one.
>
> All source code and pre-trained models will be made publicly available for further research.
>
> `Q-1:` Explanation of our technical novelty and contributions
>
> `A-1:` We highlight our technical novelty and contributions in two parts, deep unfolding framework and Transformer.
>
> (a) Deep unfolding framework
>
> Our DAUF is fundamentally different from existing deep unfolding frameworks. As analyzed in Line 5-6 and 105-106, previous unfolding methods do not estimate the degradation patterns and ill-posedness degree from the highly related CASSI system to guide the iterative learning. In contrast, we formulate a principled degradation-aware unfolding framework (DAUF) to address this issue. As mentioned in Line 74-78 and 139-142, our DAUF estimates parameters from the compressed measurement and physical mask in the CASSI system. Then the parameters, which capture critical cues of CASSI degradation patterns and ill-posedness degree, are fed into each iteration to contextually scale the linear projection and provide noise level information for the denoising network. The ablation study in the third subpart of Sec. 3.4 and Tab. 2(c) shows that our DAUF significantly outperforms previous unfolding frameworks, i.e., DNU, GAP, and ADMM by 2.59, 1.69, and 1.63 dB. This evidence demonstrates the impressive effectiveness and contribution of our DAUF. Besides, As shown in Fig.1 and Tab.1, our DAUHST outperforms previous deep unfolding methods by over 4 dB.
>
> (b) Transformer
>
> Our HST is fundamentally different from previous Transformers including those that have been used in other low-level vision tasks. As analyzed in Line 68 - 71, the computational complexity of Global Transformer is quadratic to the spatial size of the input map. This burden is non-trivial and sometimes unaffordable. The receptive fields of local Transformer are limited within position-specific windows. To tackle these problems, we propose HS-MSA that simultaneously develops two branches, i.e., a local branch capturing local content inside patches and a non-local branch modeling long-range dependencies through shuffle operation. Our HS-MSA can enjoy larger receptive fields and relatively cheaper computational costs. In Tab. 2(b) and the second subpart of Sec. 3.4, our HS-MSA outperforms G-MSA and SW-MSA by 0.42 and 0.30 dB. The improvements of our HS-MSA will enlarge when the number of stages increases.
>
> In addition, our HS-MSA is different from S-MSA of MST in
>
> (i) Motivation. S-MSA is motivated by the fact that HSI representations are spatially sparse while spectrally self-similar. Thus, MST captures the inter-dependencies in spectral dimension while circumventing the HSI spatial sparsity nature. Our HS-MSA is proposed to simultaneously model local and non-local interactions in spatial dimension, which alleviates the limitations and combines the advantages of local and global MSA. Hence, our HS-MSA fits the HSI spatial sparsity nature better than S-MSA.
>
> (ii) Technique. S-MSA treats each single-channel feature map $\in \mathbb{R}^{H\times W}$ as a token and computes self-attention in channel wise. Our HS-MSA treats each pixel vector $\in \mathbb{R}^{C}$ as a token and calculates self-attention in spatial wise. Therefore, our HS-MSA can capture more fine-grained pixel-level HSI self-similarity. Besides, the position embedding of S-MSA is generated by CNN while our position embedding is set as learnable parameters.
>
> (iii) Performance. In Tab. 2(b), our HS-MSA achieves 0.23 dB improvement over MST. Nonetheless, this margin is just for one stage and without position embedding. As shown in the following table, we conduct an ablation study by plugging MST and HST into our DAUF and increase the number of stages. PSNR / SSIM are reported. As can be seen that the improvement of HST over MST enlarges as the number of stages increases. In particular, our DAUHST-9stg significantly outperforms DAUMST-9stg by 1.35 dB, suggesting the advantage and effectiveness of our HST.
>
> | Stage | 1 | 3 | 5 | 7 | 9 |
> | - | - | - | - | - | - |
> | DAUMST | 34.01 / 0.927 | 36.47 / 0.954 | 36.89 / 0.958 | 36.95 / 0.959 | 37.01 / 0.959 |
> | DAUHST | 34.36 / 0.932 | 37.21 / 0.959 | 37.75 / 0.962 | 38.20 / 0.968 | 38.36 / 0.967 |

---

> ### Author Response · Authors · 2022-08-05
> **Further discussions with Reviewer 6Fb8**
>
> Dear Reviewer 6Fb8,
>
> We thank you for your previous review time and valuable comments. We have provided corresponding responses and results, which we believe have covered your concerns about the technical novelty, comparisons with U-Transformer, ablation for the choice of U-shaped architecture, limitation, and ablation of robustness evaluation.
>
> We hope to further discuss with you whether your concerns have been addressed or not. If you still have any unclear parts of our work, please let us know. Thanks.
>
> Best regards,
>
> Authors

---

### Official Review · Reviewer_BuNS · 2022-07-11

**Rating:** 8
**Confidence:** 5
**Soundness:** 4 excellent
**Presentation:** 4 excellent
**Contribution:** 4 excellent

**Summary:**

This work proposes a novel unfolding Transformer-based algorithm for hyperspectral image reconstruction in CASSI system. There are two key components of the proposed method:

(i) degradation-aware unfolding framework perceiving the CASSI degradation pattern and degree of ill-posedness, and

(ii) half-shuffle Transformer capturing local contents and non-local dependencies jointly.

Half-shuffle Transformer is plugged into degradation-aware unfolding framework to establish the proposed degradation-aware unfolding half-shuffle Transformer.

**Questions:**

In Table 1, I am not sure why the FLOPS of BIRNAT is so large? Is it because BIRNAT employs an RNN architecture?

**Limitations:**

The authors have discussed the limitations and social impacts in the supplementary material.
This work does not have any foreseeable negative social impacts.

**Strengths And Weaknesses:**

Strengths:

(i) This work proposes the first Transformer-based unfolding method for HSI reconstruction in snapshot compressive imaging.

(ii) The idea is novel and interesting. The degradation-aware unfolding framework changes the general unfolding paradigm of previous works (e.g., ADMM-Net, GAP-Net) by estimating parameters from CASSI system to adapt the iterative learning. Besides, the elaborately customized half-shuffle Transformer combines the advantages of local and global Transformers while using moderate Params and FLOPS.

(iii) The performance is pretty impressive. DAUHST outperforms SOTA algorithms by large margins while using less Params and FLOPS. In particular, DAUHST achieves over 4dB improvements than previous unfolding methods.

(iv) The experiments are comprehensive and the ablation study is well-designed. The comparisons in Table 1, Figure 4, and Figure 5 including 14 SOTA methods, which is very convincing. The ablation studies in Table 2 sufficiently evaluate the effectiveness of the proposed techniques. The in-depth visual analysis in Figure 6 reveals some important insights of unfolding frameworks for SCI.

(v) The writing is good and easy to follow. Especially in Section 2.1. The mathematical notation and formulation are clear, rigorous, and neatly arranged. In addition, the presentation is well-dressed.

(vi) The reproducibility can be guaranteed. I have check the submitted codes and models, and run the demo. The results in Table 1 of the main paper can be reproduced. The implementation details are described in Section 3.1. Meanwhile, the authors have claimed that they will release the code and models to the public. I believe this work can significantly benefit the community.

Weakness:

(i) The authors provide visual comparisons of DAUHST, HDNet, and BIRNAT in the supplementary but the main paper lacks these important comparisons. Why not move these results to the main paper?

(ii) In Line 156 of the main paper, the mentioned reshape operation of R is unclear and should be explained.

(iii) In Line 98 of the paper, the “vectorized” and “shifted” operations need more explanations.

(iv) It would be better to evaluate the generality and extensibility of DAUF. For instance, replacing the HST by the same CNN of GAP-Net and ADMM-Net. This ablation can show whether the proposed DAUF fits CNN-based denoising networks or not.

---

> ### Author Response · Authors · 2022-07-29
> **Response to Reviewer BuNS**
>
> Thanks for your valuable comments, now we answer your questions one by one.
>
> All source code and pre-trained models will be made publicly available for further research.
>
> `Q-1:` Move the visual results of DAUHST, HDNet, and BIRNAT from the supplementary to the main paper.
>
> `A-1:` Due to space limitation, we present these comparisons in the supplementary. Thanks for your advice, we will add these results to the main paper in the revision.
>
>
> `Q-2:` Explanation of the reshape operation of $\mathbf{R}$ in Line 156 of the main paper.
>
> `A-2:` $\mathbf{R}$ is reshaped from $\mathbb{R}^{H \times (W+d(N_\lambda - 1)) \times N_\lambda}$ to $\mathbb{R}^{H (W+d(N_\lambda - 1)) N_\lambda}$
>
>
> `Q-3:` Explanation of the vectorized and shifted operations in Line 98 of the main paper.
>
> `A-3:` Please refer to Sec.1 of the supplementary for the mathematical model of CASSI.
>
> (a) Vectorized operation
>
> The vectorized operation is detailed in Line 28 - 33 of the supplementary. It concatenates all the columns of a matrix as one single vector. Specifically, define $\tilde{\mathbf{X}} \in \mathbb{R}^{H\times (W+d(N_\lambda - 1))\times N_\lambda}$ as the shifted version of the original HSI signal $\mathbf{X} \in \mathbb{R}^{H\times W \times N_\lambda}$ . Denote the vectorized operation as $\text{vec( )}$. Then $\mathbf{x}$ in Line 98 can be formulated as
>
> $\mathbf{x} = \text{vec}([\tilde{\mathbf{x}}^{(1)}, ..., \tilde{\mathbf{x}}^{(N_\lambda )}])$,
>
> where $\tilde{\mathbf{x}}^{(n_{\lambda} )} = \text{vec}(\tilde{\mathbf{X}}(:, :, n_\lambda ))$ and $[..., ...]$ indicates the concatenating operation
>
> (b) Shifting operation
>
> The shifting is caused by the disperser in CASSI. The shifted operation is detailed in Line 13 - 21, and Eq (2) of the supplementary. After passing through a disperser, the 3D HSI cube $\mathbf{X}' \in \mathbb{R}^{H\times W \times N_\lambda}$ becomes tilted and could be considered as sheared along the $y$-axis. Define $\mathbf{X}'' \in \mathbb{R}^{H\times (W+d(N_\lambda - 1))\times N_\lambda}$ as the tilted cube, and $\lambda_c$ as the reference wavelength, i.e., $\mathbf{X}'[:, :, n_{\lambda_c} ]$ is not sheared along the $y$-axis. Then the dispersion is formulated as
>
> $\mathbf{X}''(u, v, n_{\lambda}) = \mathbf{X}'(x, y+d(\lambda_n - \lambda_c), n_{\lambda})$ ,
>
> where $(u, v)$ indicates the coordinate system on the detector plane, $\lambda_n$ denotes the wavelength of the $n_{\lambda}$-th spectral channel, $d$ represents the shifting step, and $d(\lambda_n - \lambda_c)$ signifies the spatial shifting for the $n_{\lambda}$-th channel on $\mathbf{X}'$ . Thus, the shifted version $\tilde{\mathbf{X}}$ of the original HSI cube $\mathbf{X}$ can be similarly formulated as
>
> $\tilde{\mathbf{X}} (u, v, n_{\lambda}) = \mathbf{X}(x, y+d(\lambda_n - \lambda_c), n_{\lambda})$ .
>
> `Q-4:` Comparison of DAUF, GAP, and ADMM on CNN-based methods
>
> `A-4:` Thanks for your advice. We conduct the ablation study as you recommend in the following table. Three-stage architecture is adopted. For each stage, the same U-shaped CNN as GAP-Net and ADMM-Net is adopted. It can be observed that our DAUF significantly outperforms GAP and ADMM by 2.75 and 2.32 dB in PSNR, 0.046 and 0.038 in SSIM. These results demonstrate the impressive effectiveness and promising generalization ability of our DAUF.
>
> | Method | GAP | ADMM | DAUF |
> | ---------- | ------ | --------- | -------- |
> | PSNR | 32.06 | 32.49 | 34.81 |
> | SSIM | 0.892 | 0.900 | 0.938 |
>
> `Q-5:` Question about the FLOPS of BIRNAT.
>
> `A-5:` Yes, you are right. BIRNAT suffers from enormous FLOPS because it adopts an RNN architecture that recurrently processes images for many times (28 times for HSIs in the official implementation of BIRNAT).

---

> > ### Comment · Reviewer_BuNS · 2022-08-07
> > **Response to Authors' Rebuttal**
> >
> > I have read the author's feedback. My confusion regarding some details and the concerns raised by other reviewers are well-addressed. The author's responses are well organized and experimentally supported, which is convincing. Thanks for their effort. Here is my response to the author’s rebuttal.
> >
> > (i) The motivations and novelties are excellent.
> >
> > I have investigated snapshot compressive imaging for decades. To the best of my knowledge, this is the first Transformer-based deep unfolding method in snapshot compressive imaging. This work contributes a novel deep unfolding framework and an interesting Transformer for snapshot compressive imaging. The proposed DAUF can perceive CASSI degradation to adjust the iterative learning. The proposed HS-MSA computes spatial self-attention through shuffle operation, capturing long-range dependencies while costing cheaper.
> >
> > (ii) The improvements over state-of-the-art methods are very significant.
> >
> > As reported in Table 1, the proposed DAUHST dramatically outperforms previous deep unfolding methods by over 4 dB and state-of-the-art method by 0.78 dB.
> >
> > (iii) The reproducibility can be ensured.
> >
> > I have checked the submitted code and models. The reported results can be exactly reproduced. I believe the subsequent open-source works will significantly benefit the community and further research of snapshot compressive imaging.
> >
> > Bearing the above considerations in mind, I highly recommend this paper to be accepted.

---

> > > ### Author Response · Authors · 2022-08-09
> > > **Thank Reviewer BuNS for highly praising our work**
> > >
> > > Dear Reviewer BuNS,
> > >
> > > Thanks for discussing with us and agreeing that our response well addresses your concerns. Thank you for highly praising the $\textbf{novelty}$, $\textbf{motivation}$, $\textbf{contribution}$, and $\textbf{reproducibility}$ of our work. We will release all the codes and pre-trained models to the public for further research.
> > >
> > > Best regards,
> > >
> > > Authors

---

> ### Author Response · Authors · 2022-08-05
> **Further discussions with Reviewer BuNS**
>
> Dear Reviewer BuNS,
>
> We thank you for your previous review time and valuable comments. We have provided corresponding responses and results, which we believe have covered your concerns about the placement of visual results, reshape operation of $\mathbf{R}$, the vectorized and shifted operations, generality of DAUF, and FLOPS of BIRNAT.
>
> We hope to further discuss with you whether your concerns have been addressed or not. If you still have any unclear parts of our work, please let us know. Thanks.
>
> Best regards,
>
> Authors

---

### Official Review · Reviewer_Jov1 · 2022-07-11

**Rating:** 4
**Confidence:** 5
**Soundness:** 2 fair
**Presentation:** 3 good
**Contribution:** 2 fair

**Summary:**

This paper proposes a degradation-aware unfolding framework (DAUF) where the parameters are estimated by the input image and physical mask, which can serve as a guidance for each iteration stage towards the final output. Among each iteration stage, two parts, e.g., a linear projection and denoising network, perform successively based on maximum a posteriori (MAP). Specifically, the denoising network builds on Transformer-type framework replacing the vanilla self-attention mechanism with two branches, local and non-local branch, to extract the local and long-dependency representations.

**Questions:**

1, For the input of denoising network, beta_k is stretched to concatenate with x_k following the Eq. 12, and in Tab 2(d) the learnable alpha is studied. I wonder if beta and alpha can be both set as learnable parameters and achieve a compatible result, i.e., learnable beta is also stretched as the original method do or just be added into x_k . In that case, the method would not be degradation-aware.
2, In 3.4, HS-MSA is remove in the first two subparts. Is it substituted by identity operation? DAUHST-3stg is used to perform ablation studies except for self-attention mechanism (DAUHST-1stg). Is it because the memory bottlenecks of global self-attention?
3, In Eq. 3, auxiliary variable z is introduced to replace x in the image prior term R(*), but this replacement occurs in the data term in Eq4. of [1]. What is the difference?
4, It would be better to provide the comparison of complexity for attention alternatives.
5, The DAUF works in an iterative manner. Is the same HST used or not for different stages? Is there any difference between training and test phase? For example, should the test input go though all stages?
[1] Deep Unfolding Network for Image Super-Resolution, CVPR 2020.


**Limitations:**

The method works in an iterative manner where there is a Transformer-type HST in each stage, which may lead to a long training and inference time. In addition, the non-local branch captures relatively long-range dependencies but not globally.

**Strengths And Weaknesses:**

Strengths:
The paper is clearly written and the logic flow is easy to follow. The method sufficiently utilizes the potential prior knowledge in the input image and the mask to guide the reconstruction process. For the denoising sublayer, local window based and shuffle based self-attention are leveraged to capture local and long-range dependencies. Overall, the approach achieves over 4 dB higher performance than SOTA deep unfolding based methods, and 0.78 dB than SOTA.

Weaknesses:
In my opinion, the authors combine three approaches to establish their method [1,2,3]. To be specific, the authors of [1] also explore a deep unfolding network based on MAP and HQS and then get two parameters, alpha and beta, to generate x_k and z_k iteratively. With respect to the two-branch attention paradigm, the local attention based on windows and long-range attention based on shuffle operation have already put forward in [2] but in successive manner instead of parallel. For the flow of transformer, this paper follows most parts of [3], e.g., FFN part and FPN-style connection. To sum up, I doubt the novelty of the method. If authors are inspired by these papers, it would be better to include them in the reference. If I am wrong, please correct me.

[1] Deep Unfolding Network for Image Super-Resolution, CVPR 2020.
[2] CAT: Cross Attention in Vision Transformer, arxiv 2021.
[3] Mask-Guided Spectral-Wise Transformer for Efficient Hyperspectral Image Reconstruction, CVPR 2022.

---

> ### Author Response · Authors · 2022-07-30
> **Response to Reviewer Jov1 - Part 4**
>
> `Q-11:` Does the non-local branch capture relatively long-range or global dependencies?
>
> `A-11:` We claim that the non-local branch captures global dependencies. See Fig. 3 (d). For each patch, after the shuffle operation, all patches in the global area are extracted a token to this patch for calculating self-attention. In other words, the non-local branch calculates self-attention inside each patch with tokens that come from all patches in the whole image. Besides, the local branch computes self-attention with all tokens inside a patch. Thus, the interactions between any two tokens in the whole image can be established and the global dependencies can be captured.

---

> ### Author Response · Authors · 2022-07-30
> **Response to Reviewer Jov1 - Part 3**
>
> `Q-4:` Ablation study of setting both $\alpha$ and $\beta$ as learnable parameters
>
> `A-4:` We conduct the ablation as you recommend in the following table. We adopt DAUHST-3stg as Baseline-3 but $\alpha$ is set as learnable parameters and $\beta$ is not used. If $\alpha$ and $\beta$ are both learnable parameters, the improvements are marginal, i.e., 0.02 dB when $\beta_k$ adds with $x_k$ and 0.13 dB when $\beta_k$ concatenates with $x_k$ . However, our DAUF gains by 0.72 dB, which surpasses the degradation-unaware schemes by over 0.59 dB. This evidence suggests that capturing CASSI degradation patterns and ill-posedness degree is informative for the iterative learning. Thanks for your advice, we will add this ablation in the revision
>
> | Method | $\beta_k$ with $x_k$ | PSNR | SSIM | Params (M) | FLOPS (G) |
> | - | - | - | - | - | - |
> | Baseline-3 | None | 36.49 | 0.952 | 2.03 | 26.23 |
> | Both Learnable | add | 36.51 | 0.952 | 2.03 | 26.23 |
> | Both Learnable | concate | 36.62 | 0.953 | 2.03 | 26.29 |
> | DAUF | concate | 37.21 | 0.959 | 2.08 | 27.17 |
>
> `Q-5:` Question about removing HS-MSA
>
> `A-5:` In Fig.3 (b), removing HS-MSA indicates that the first layer normalization, HS-MSA, and the first skip connection are all removed and replaced by an identity operation. Sorry for confusing, we clarify this point in the revision
>
> `Q-6:` Why does Tab. 2(b) adopts a one-stage architecture?
>
> `A-6:` The memory bottleneck of global self-attention is one of the reasons. In this ablation study, we focus on comparing different self-attention mechanisms. Thus, we reduce the effect of multi-stage iterative learning by adopting a single-stage structure. By the way, we are glad to share the ablation study on three-stage architecture in the following table. Baseline-4 indicates removing HS-MSA from DAUHST-3stg. Different position embedding schemes are removed for stronger comparison. Our HS-MSA outperforms G-MSA, S-MSA, and SW-MSA by 0.95, 0.78, and 0.66 dB.
>
> | Method | Baseline-4 | G-MSA | SW-MSA | S-MSA | HS-MSA |
> | - | - | - | - | - | - |
> | PSNR | 35.37 | 35.84 | 36.13 | 36.01 | 36.79 |
> | SSIM | 0.938 | 0.948 | 0.952 | 0.950 | 0.956 |
> | Params (M) | 1.11 | 1.35 | 1.35 | 1.35 | 1.35 |
> | FLOPS (G) | 18.55 | 28.91 | 26.24 | 24.68 | 27.17 |
>
> `Q-7:` Question about the auxiliary variable
>
> `A-7:` The different positions of the auxiliary variable lead to different unfolding inferences and subsequent iterative learning. We introduce $z$ into the image prior term. In the iteration, the solution $x$ is firstly handled while the auxiliary variable $z$ is subsequently solved. USRNet introduces the auxiliary variable into the data term. Then in the iteration, the auxiliary variable $z$ is firstly solved while the solution $x$ is secondly handled. Please refer to `A-1` of `Response to Reviewer Jov1 - Part 1` for more comparisons between DAUHST and USRNet.
>
> `Q-8:` Question about the comparison of complexity for attention alternatives
>
> `A-8:` In fact, we have compared the memory and computational complexity of using different self-attention mechanisms in Tab. 2(b) and the second subpart of Sec. 3.4. In Tab. 2(b), Params indicates the memory complexity while FLOPS indicates computational complexity. Besides, we also provide theoretical analysis of computational complexity in Sec. 2 of the supplementary. We are glad to list the theoretical computational complexity in the following table. Suppose the input size as $H \times W \times C$. In implementation, $H = W = 256$, $M = 8$, $C = 28$. Our HS-MSA only requires 0.89% computational costs of G-MSA but also captures global spatial interactions, showing its efficiency advantage.
>
> | Self-Attention | Computational Complexity |
> | - | - |
> | G-MSA | $4H\hat W C^2+2(H\hat W)^2C$ |
> | SW-MSA | $4H\hat W C^2+2M^2H\hat WC$ |
> | S-MSA | $4H\hat W C^2+2H\hat WC^2/N$ |
> | HS-MSA | $4H\hat W C^2+M^2H\hat WC+H^2\hat W^2C/M^2$ |
>
> `Q-9:` Questions about weights sharing, training and testing phases
>
> `A-9:` Different stages do not share parameters or weights although they use the same HST architecture because weight-sharing operation limits the representing capacity of the whole network. Note that DAUHST outperforms SOTA methods with less Params even without weight-sharing operation, see Tab. 1 of the paper. There is no difference between training and testing phases. The test input should go through all stages.
>
> `Q-10:` Comparison of training and inference time
>
> `A-10:` We admit that the performance improvement of our method comes with increasing training and testing time. However, we compare DAUHST with SOTA methods on the same GPU in the following table. The training and testing time of DAUHST is moderate while the improvement of DAUHST is significant.
>
> | Method | DNU | TSA-Net | GAP-Net | MST-L | DAUHST-3stg |
> | - | - | - | - | - | - |
> | PSNR | 30.74 | 31.46 | 33.26 | 35.18 | 37.21 |
> | SSIM | 0.863 | 0.894 | 0.917 | 0.948 | 0.959 |
> | Train (h) | 253 | 69 | 61 | 100 | 70 |
> | Ineference (ms) | 558 | 68 | 60 | 90 | 76 |

---

> ### Author Response · Authors · 2022-07-30
> **Response to Reviewer Jov1 - Part 2**
>
> `Q-2:` Comparison with CAT
>
> `A-2:` Our DAUHST is different from CAT in
>
> (a) Non-local self-attention mechanism
>
> $\textbf{CAT does not use shuffle operations to capture long-range dependencies.}$ In Sec. 3.1.2 of CAT [2], the cross-patch self-attention (CPSA) partitions each single-channel feature map into patches. Then CPSA treats each patch $\in \mathbb{R}^{N\times N}$ as a token to compute self-attention. CPSA is the same as the global multi-head self-attention (MSA) in ViT except that CPSA computes self-attention in a single channel at a time while global MSA calculates self-attention with all channels. In other words, CPSA is a depth-wise separable global MSA. Yet, this scheme limits the representing capacity and easily causes inaccurate matching because the representations and interactions of other channels are neglected when computing self-attention in a specific channel. In contrast, our HS-MSA exchanges the spatial positions of pixel vectors in different patches through shuffle operations. Then HS-MSA treats each pixel vector $\in \mathbb{R}^{C}$ as a token to calculate self-attention within each patch, see Fig. 3(d). Subsequently, an unshuffle operation is exploited to restore the original positions of tokens. By this means, HS-MSA establishes long-range dependencies
>
> (b) Network depth, memory and computational costs
>
> The global and local self-attention of CAT are cascaded in a successive manner. There are three self-attention units in a basic block, see Fig. 2 (b) in [2]. This leads to two issues. Firstly, the local contents and global interactions are not simultaneously captured. Secondly, the depth of CAT is deepened, introducing difficulties into the training process, increasing the memory and computational costs, and lowering down the model inference speed. In contrast, HS-MSA simultaneously develops two parallel branches to extract local contents and non-local interactions, without increasing the depth and costs. Thus, HST is more cost-effective, efficient, and easier to train
>
> (c) Motivation and technique
>
> CAT is proposed as a backbone for high-level vision tasks. In contrast, HST is designed as a U-shaped structure to be plugged into unfolding framework for SCI reconstruction. Besides, CAT adopts absolute position embedding and patch embedding layer while HST applies learnable position embedding
>
> (d) Experiment
>
> We adopt the two self-attention proposed by CAT, i.e., Inner-Patch Self-Attention (IPSA) and CPSA to replace HS-MSA in HST to conduct ablations in the following table. Our HS-MSA costs less Params, FLOPS, and training time but enjoys higher performance and faster inference speed, showing the advantages of HST
>
> | Method | PSNR | SSIM | Params (M) | FLOPS (G) | Train Time (h) | Test Time (ms) |
> | - | - | - | - | - | - | - |
> | IPSA + CPSA | 33.14 | 0.917 | 0.49 | 10.37 | 32  | 42  |
> | HS-MSA | 34.05 | 0.930 | 0.48 | 9.72 | 24 | 29 |
>
> We will follow your advice to cite USRNet [1] and CAT [2], and add these results in the revision
>
> `Q-3:` Comparison between HST and MST
>
> `A-3:` In fact, the FFN and FPN-style connection of HST and MST both follow previous works, such as Swin Transformer, U-former, PVT, etc. However, we do not claim this part as our contributions. We state that HS-MSA is our contribution. So we keep the same flow as MST [1] for fair comparison of self-attention mechanism to show the advantages of HST. HST is different from MST in
>
> (a) Motivation
>
> MST is motivated by the fact that HSI representations are spatially sparse while spectrally self-similar. Thus, MST captures the inter-dependencies in spectral dimension while circumventing the HSI spatial sparsity nature. Our HST is proposed to jointly model local and non-local interactions with a low computational cost in spatial dimension, which alleviates the limitations and combines the advantages of local and global MSA. Hence, our HST fits the HSI spatial sparsity nature better
>
> (b) Technique
>
> MST treats each single-channel feature map $\in \mathbb{R}^{H\times W}$ as a token and computes self-attention in channel wise. HST treats each pixel vector $\in \mathbb{R}^{C}$ as a token and calculates self-attention in spatial wise. Thus, HST can capture more fine-grained pixel-level HSI self-similarity. The position embedding of MST is produced by CNN while that of HST is additive learnable parameters
>
> (c) Performance
>
> In Tab. 2(b), HS-MSA yields 0.23 dB improvements over S-MSA. Yet, this margin is just for one stage and without position embedding. In the following table, we plug MST and HST into DAUF and increase the number of stages. PSRN / SSIM are reported. The improvements of HST over MST enlarges as the number of stages increases. In particular, DAUHST-9stg outperforms DAUMST-9stg by 1.35 dB
>
> | Stage | 1 | 3 | 5 | 7 | 9 |
> | - | - | - | - | - | - |
> | DAUMST | 34.01 / 0.927 | 36.47 / 0.954 | 36.89 / 0.958 | 36.95 / 0.959 | 37.01 / 0.959 |
> | DAUHST | 34.36 / 0.932 | 37.21 / 0.959 | 37.75 / 0.962 | 38.20 / 0.968 | 38.36 / 0.967 |

---

> ### Author Response · Authors · 2022-07-30
> **Response to Reviewer Jov1 - Part 1**
>
> Thanks for your valuable comments
>
> Code and models will be made public
>
> We argue that our DAUHST is fundamentally different from USRNet [1], CAT [2], and MST [3]
>
> `Q-1:` Comparison with USRNet
>
> `A-1:` Firstly, using MAP theory to build energy function and HQS algorithm to obtain an unfolding inference are two common strategies in deep unfolding methods [4,5,6]. These methods all generate $x_k$ , $z_k$ in an iterative manner. These are not the contributions of us or USRNet. Our DAUHST and USRNet are different in
>
> (a) Research topic, degradation pattern, and motivation
>
> USRNet is designed for super-resolution (SR). The degradation in low-resolution image is caused by a blur kernel, downsampling, and an additive noise. In contrast, our DAUHST is designed for snapshot compressive imaging (SCI) reconstruction. The degradation of SCI is mainly caused by mask-modulation, dispersion, and integration-compression. The degradation patterns and mathematical models of SR and SCI are different. Since using MAP theory for deep unfolding methods requires task-specific mathematical degradation models, the established energy function and subsequent formulation of USRNet and DAUHST, especially the closed-form solutions are fundamentally different. Please compare Eq. (1) - (7) in [1] and Eq. (1) - (10) in our paper. USRNet adopts a very sophisticated SR-specific closed-form solution in Eq. (7) of [1] to update $z_k$ . In contrast, we formulate a computationally efficient SCI-specific closed-form formulation based on the characteristic of sensing matrix in Eq. (10) of our paper, which can update $x_k$ in one shot. All these lead to the large variance in the iterative learning, including the formulation, input, and output of each iteration
>
> (b) The estimation method, implication, and function of the estimated parameters
>
> See the third subpart of Sec. 3.3 in [1], the hyperparameter module of USRNet directly adopts the noise level $\mathbf{σ}$ and scale factor $\mathbf{s}$ as inputs. Plus, the blur kernel $\mathbf{k}$ is directly applied in the subsequent iterative learning. In real practice, these degradation parameters are usually not given in advance and thus need manual tweaking. As a result, USRNet suffers from poor generality. In application, usually only the degraded image is given. Yet, the hyperparameter module does not take the low-resolution image as input. Thus, the estimated parameters do not capture the feature and degradation pattern from the low-resolution image but only represent the manually tweaked parameter $\mathbf{σ}$ and $\mathbf{s}$ to function as a ‘slide bar’ to control the input and output of each iteration. In contrast, our estimator directly operates and extracts information from the compressed measurement and physical mask. Hence, the estimated parameters can capture the feature and CASSI degradation pattern from the degraded image. Besides, the mask is usually given in advance. Thus, DAUHST does not need manual tweaking and enjoys better generalization ability
>
> (c) Technique
>
> USRNet is based on CNN, showing limitations in capturing long-range dependencies. DAUHST based on Transformer addresses this issue. Our HS-MSA jointly develops two branches. The local branch extracts local contents and the non-local branch models long-range interactions through shuffle operation. Besides, all the networks in each iteration of USRNet share the same weights. This forces a single model to fit different image processing functions of different iterations. Yet, the representing ability of a single model is limited. Thus, this weights-sharing operation limits the progressive image restoration performance, especially in SCI reconstruction that deals with sophisticated and severely ill-posed CASSI degradation. In contrast, DAUHST does not share weights of networks. Thus, DAUHST enjoys powerful representing ability to simulate the progressive SCI reconstruction across stages
>
> (d) Experiment
>
> We perform ablations to compare with USRNet in the following table. USRHST indicates replacing U-Net in USRNet with our HST. DAUHST-share means sharing weights across stages like USRNet. Three-stage architecture is adopted. DAUHST-share outperforms USRHST by 1.06 dB, showing our DAUF is more suitable for SCI. USRHST surpasses USRNet by a large margin with less Params and FLOPS, which stems from the capacity of HST to model long-range dependencies. DAUHST outperforms DAUHST-share by 0.48 dB, showing that weights sharing operation sacrifices the representation ability
>
> | Method| PSNR | SSIM | Params (M) | FLOPS (G) |
> | - | - | - | - | - |
> | USRNet | 35.13 | 0.942 | 2.32 | 39.39 |
> | USRHST | 35.67 | 0.947 | 0.69 | 26.30 |
> | DAUHST-share | 36.73 | 0.955 | 0.73 | 27.17 |
> | DAUHST | 37.21 | 0.959 | 2.08 | 27.17 |
>
> `Reference`
>
> [4] Nonlinear image recovery with half quadratic regularization. In TIP 1995
>
> [5] Fast Image Deconvolution using Hyper-Laplacian Priors. In NeurIPS 2009
>
> [6] Multi-Scale Patch-Based Image Restoration. In TIP 2015

---

> ### Author Response · Authors · 2022-08-05
> **Further discussions with Reviewer Jov1**
>
> Dear Reviewer Jov1,
>
> We thank you for your previous review time and valuable comments. We have provided corresponding responses and results, which we believe have covered your concerns about the novelty, motivation, ablation of $\alpha$ and $\beta$, removing HS-MSA, ablation of self-attention with three-stage architecture, position of the auxiliary variable, comparison of complexity for attention alternatives, weight sharing operation, difference between training and testing, limitation, and receptive fields of non-local branch.
>
> We hope to further discuss with you whether your concerns have been addressed or not. If you still have any unclear parts of our work, please let us know. Thanks.
>
> Best regards,
>
> Authors

---

> ### Author Response · Authors · 2022-08-08
> **Waiting for reply of Reviewer Jov1**
>
> Dear Reviewer Jov1,
>
> We thank you for your previous review time and valuable comments. Until now, the other three Reviewers have replied to us that their concerns including the $\textbf{motivation}$, $\textbf{contribution}$, and limitation of our work are mostly addressed. We also believe our response to you has covered your concerns. We hope to further discuss with you whether your concerns have been addressed or not. If you still have any unclear parts of our work, please let us know. Thanks.
>
> The author-reviewer discussion period will end on August 9. We are still waiting for your reply.
>
> Best regards,
>
> Authors

---

> > ### Comment · Reviewer_Jov1 · 2022-08-09
> > **response to the authors**
> >
> > Thank you so much for the response.
> >
> > Although I've read the feedback from the authors, my main concern remains that the proposed method's novelty and gain are limited. For instance, both parameters and FLOPs are not better than USR.
> >
> > Moreover, the authors mention in the response that "USRNet is based on CNN, showing limitations in capturing long-range dependencies. DAUHST based on Transformer addresses this issue. Our HS-MSA jointly develops two branches", which also demonstrates the proposed method as a combination of existing work. Therefore, the novelty raised by the authors is overclaimed.

---

> > > ### Author Response · Authors · 2022-08-09
> > > **Discussions with Reviewer Jov1**
> > >
> > > Thanks for your response. Now we answer your questions.
> > >
> > >
> > > $\textbf{Firstly}$, we have provided the quantitative comparisons between USRNet and DAUHST in `A-1` (d) of `Response to Reviewer Jov1 - Part 1` . We also list the results here in the following table. DAUHST-share means using weights sharing operation like USRNet. It can be observed that our DAUHST significantly outperforms USRNet by $\textbf{2.08 dB}$ while only requiring $\textbf{89.7}$% (2.08 / 2.32) Params and $\textbf{69.0}$% (27.17 / 39.39) FLOPS. Our DAUHST-share significantly surpasses USRNet by $\textbf{1.60 dB}$ while only costing $\textbf{31.5}$% Params and $\textbf{69.0}$% FLOPS. All these results demonstrate the effectiveness superiority and efficiency advantage of our DAUHST over USRNet.
> > >
> > > | Method | PSNR | SSIM | Params (M) | FLOPS (G) |
> > > | - | - | - | - | - |
> > > | USRNet | 35.13 | 0.942 | 2.32 | 39.39 |
> > > | DAUHST-share | 36.73 | 0.955 | 0.73 | 27.17 |
> > > | DAUHST | 37.21 | 0.959 | 2.08 | 27.17 |
> > >
> > > $\textbf{Secondly}$, we do not understand the logical causal relationship between our statement that “USRNet is based on CNN, showing limitations in capturing long-range dependencies. DAUHST based on Transformer addresses this issue. Our HS-MSA jointly develops two branches” and your conclusion that our method is a combination of existing work. In fact, we have already stressed the differences between our DAUHST and the three works you mentioned (i.e., USRNet, CAT, and MST.) in ` Response to Reviewer Jov1 - Part 1` and ` Response to Reviewer Jov1 - Part 2`. They are fundamentally different in $\textbf{motivation}$, $\textbf{contribution}$, $\textbf{technique}$, etc. We also conduct comprehensive experiments to show the significant advantages of our method.
> > >
> > > We highlight our technical novelty and contributions in two parts, $\textbf{deep unfolding framework}$ and $\textbf{Transformer}$. Please refer to `A-1` of ` Response to Reviewer 6Fb8 - Part 1` for details. Besides, the other reviewers’ concerns about our $\textbf{motivation}$ and $\textbf{contribution}$ have been addressed.
> > >
> > > We hope to further discuss with you whether your concerns have been addressed or not. If you still have any unclear parts of our work, please let us know. Thanks.
> > >
> > > Best regards,
> > >
> > > Authors

---

### Official Review · Reviewer_nC4p · 2022-07-11

**Rating:** 5
**Confidence:** 4
**Soundness:** 3 good
**Presentation:** 3 good
**Contribution:** 3 good

**Summary:**

This paper introduces a hyperspectral image reconstruction method based on a deep unfolding method and a Half-shuffle Transformer (HST). The deep unfolding method estimates parameters from the image and uses them to direct the iterative learning. The HST captures both local contents and non-local dependencies with relatively low computation cost. The proposed method was evaluated on both synthetic and real image datasets and demonstrated excellent performance over a few other methods. Ablation studies were also provided to show the contribution of key components in the proposed method.

**Questions:**

Please address the comments in the Strengths And Weaknesses.

**Limitations:**

The spectral ranges covered by the images in the adopted synthetic datasets are narrow. Please consider experiments with more infrared spectrum.

**Strengths And Weaknesses:**

This is a solid paper. Each proposed component in the whole method structure seems to be working, and when putting together, they produce the best hyperspectral image reconstruction performance reported so far. However, I also have several concerns about this paper.

First, the proposed DAUHST method has its novelty in combining transformer and deep unfolding for HSI restoration. However, the contribution can only be considered incremental since a similar idea (transformer as prior + unfolding) has been proposed for hyperspectral image super-resolution in November 2021, which is closely related to the reconstruction or restoration task: Learning A 3D-CNN and Transformer Prior for Hyperspectral Image Super-Resolution. Furthermore, both deep unfolding and transformer have been adopted separately for hyperspectral image processing. Take the self-attention HS-MSA module in the proposed Half-Shuffle Transformer (HST) as an example, it can be considered an extension of the S-MSA in MST-L [81]. Table 2 (b) also shows that the performance improvement of HS-MSA over S-MSA is marginal.

Second, the introduction section hasn't clearly described the motivation of the degradation-aware unfolding framework. It simply says previous methods do not estimate CASSI degradation patterns and ill-posedness degree. Please clearly define what are CASSI degradation patterns and ill-posedness degree and why they are important to the estimation process.

The method description also needs to be improved as well. In Figure 2, the parameter estimator takes the compressed measurement and the sensing of the CASSI system as inputs and produces \alpha and \beta. How is the estimation done? The details are missing. In lines 127-129, \mu and \tau are iteration-specific parameters. How their values are determined in each iteration?

Table 1 shows that the 9 stages DAUHST produces the best results. An analysis of the influence of the number of stages on the final performance shall be given.

---

> ### Author Response · Authors · 2022-07-30
> **Response to Reviewer nC4p - Part 3**
>
> `Q-7:` The influence of the number of stages
>
> `A-7:` In fact, we have conducted this ablation in Sec. 4.1 and Tab. 1(a) of the supplementary. We also list the results here in the following table. The performance improves when we gradually increase the stage number. We notice that a 3-stage DAUHST can achieve a very impressive PSNR result of 37.21 dB.
>
> | Stage | 1 | 3 | 5 | 7 | 9 |
> | - | - | - | - | - | - |
> | PSNR | 34.36 | 37.21 | 37.75 | 38.20 | 38.36 |
> | SSIM| 0.932 | 0.959 | 0.962 | 0.968 | 0.967 |
> | Params (M) | 0.73 | 2.08 | 3.44 | 4.79 | 6.15 |
> | FLOPS (G) | 9.72 | 27.17 | 44.61 | 62.05 | 79.50 |
>
> `Q-8:` The spectral ranges covered by the images in the synthetic datasets are narrow. Experiments with more infrared spectrum.
>
> `A-8:` (a) We use the same datasets to evaluate the SCI reconstruction performance as previous snapshot compressive imaging reconstruction methods such as GAP-Net, DGSMP, TSA-Net, and PnP-DIP-SCI for fair comparison.
>
> (b) We are glad to conduct the experiment as you recommend. We adopt the Botswana dataset (https://www.ehu.eus/ccwintco/index.php/Hyperspectral_Remote_Sensing_Scenes) that contains more infrared spectra as you suggest. The spectral wavelength range of the Botswana dataset is 400 nm - 2500 nm.  We uniformly select 28 out of 145 spectra and randomly crop patches with a spatial size of 256×256 from the Botswana dataset as training samples. The results are listed in the following table. Our DAUHST still outperforms other SOTA methods while requiring the least computational cost. These results demonstrate the cost-effective advantage and generalization ability of our DAUHST.
>
> | Method| GAP-Net | TSA-Net | MST-L | DAUHST-3stg |
> | - | - | - | - | - |
> | PSNR | 29.36 | 31.52 | 31.78 | 32.45 |
> | SSIM| 0.705 | 0.735 | 0.737 | 0.751 |
> | Params (M) | 4.27 | 44.25 | 2.03 | 2.08 |
> | FLOPS (G) | 78.58 | 110.06 | 28.15 | 27.17 |

---

> ### Author Response · Authors · 2022-07-30
> **Response to Reviewer nC4p - Part 2**
>
> `Q-3:` Comparison between HS-MSA and S-MSA
>
> `A-3:` HS-MSA is fundamentally different from S-MSA in
>
> (a) Motivation
>
> S-MSA is motivated by the fact that HSI representations are spatially sparse while spectrally self-similar. Thus, MST captures the inter-dependencies in spectral dimension while circumventing the HSI spatial sparsity nature. HS-MSA is proposed to jointly model local and non-local interactions with a low computational cost in spatial dimension, which alleviates the limitations and combines the advantages of local and global MSA. Hence, our HS-MSA fits the HSI spatial sparsity nature better
>
> (b) Technique
>
> S-MSA treats each single-channel feature map $\in \mathbb{R}^{H\times W}$ as a token and computes self-attention in channel wise. HS-MSA treats each pixel vector $\in \mathbb{R}^{C}$ as a token and calculates self-attention in spatial wise. Thus, HS-MSA can capture more fine-grained pixel-level HSI self-similarity
>
> (c) Experiment
>
> In Tab. 2(b), HS-MSA is 0.23 dB higher than S-MSA. Yet, this margin is just for one stage and without position embedding. In the following table, we plug MST and HST into DAUF and increase the number of stages. PSRN / SSIM are reported. The improvements of HS-MSA over S-MSA enlarge as the number of stages increases. In particular, DAUHST-9stg outperforms DAUMST-9stg by 1.35 dB
>
> | Stage | 1 | 3 | 5 | 7 | 9 |
> | - | - | - | - | - | - |
> | DAUMST | 34.01 / 0.927 | 36.47 / 0.954 | 36.89 / 0.958 | 36.95 / 0.959 | 37.01 / 0.959 |
> | DAUHST | 34.36 / 0.932 | 37.21 / 0.959 | 37.75 / 0.962 | 38.20 / 0.968 | 38.36 / 0.967 |
>
> `Q-4:`  What are CASSI degradation patterns and ill-posedness degree? Why are they important?
>
> `A-4:` (a) See the measurement and GT HSIs in Fig. 4 of the paper. The CASSI degradation patterns and ill-posedness degree are that the original HSIs are compressed from a 3D cube to a single 2D image, the scene is severely distorted and blurred, and many artifacts are introduced into the measurement. In Line 34 -38, this degradation stems from the CASSI process that uses a coded aperture and a disperser to modulate the spectral signals at different wavelengths, and then mixes all modulated signal to generate a 2D compressed measurement
>
> (b) The subsequent iterative learning is based on MAP theory that is highly related to the CASSI degradation model, see Eq. (2) of the  paper. Thus, the two main phases unfolded from the MAP-based energy function Eq. (2) in each iteration, i.e., linear projection that requires scale factor in Eq. (10) and denoiser prior that requires noise level information in Eq. (11), are also closely correlated to the CASSI degradation pattern and ill-posedness degree. Yet, previous unfolding methods naively set these two parameters as constants or learnable parameters, which neglects the connection between CASSI system and the iterative learning. To tackle this issue, DAUF estimates the two parameters from the physical mask and compressed measurement. Then the parameters capturing key cues of CASSI degradation patterns and ill-posedness degree are exploited to adaptively scale the linear projection and provide noise level information for the denoiser prior. As shown in Tab. 2(c), our DAUF outperforms DNU, ADMM, and GAP by 2.59, 1.69, and 1.63 dB. As shown in Fig.1, DAUHST significantly outperforms previous deep unfolding methods by over 4 dB
>
> `Q-5:` Details about the parameter estimation
>
> `A-5:` Instead of using hand-crafted formulation or image prior that needs manual tweaking and suffers from poor generality, we adopt a learnable estimator to implicitly and automatically estimate the informative parameters from the measurement and sensing mask. The details of the estimator is described in Line 138 -142. The shifted measurement concatenated with the sensing matrix undergoes a conv1x1, a strided conv3x3, a global average pooling, and three fully connected layers to obtain the parameters. In other words, $\alpha$ and $\beta$ are estimated end-to-end by passing the measurement and sensing matrix through the estimator. You can also refer to Line 338 - 346 of code `DAUHST.py` in the supplementary for implementation
>
> `Q-6:` What determines $\mu$ and $\tau$?
>
> `A-6:` In Line 109 - 110, $\tau$ is a hyperparameter balancing the importance between data term and prior term. In Line 113, $\mu$ is a penalty parameter that forces x and z to approach the same fixed point. $\mu$ and $\tau$ are both hyperparameters that need to be set. Some previous unfolding frameworks such as GAP and ADMM set them as constants (=1). Some (e.g., DNU) set them as learnable parameters. In contrast, our DAUF set them as informative parameters that are estimated from the CASSI system. In Line 132 - 133, we set $\alpha_k = \mu_k$ and $\beta_k = \mu_k / \tau_k$ . $\alpha_k$ and $\beta_k$ are estimated from the physical mask and compressed measurement. Then $\mu_k$ and $\tau_k$ are determined by $\alpha_k$ and $\beta_k$ , i.e., $\mu_k = \alpha_k$ and $\tau_k = \alpha_k / \beta_k$

---

> ### Author Response · Authors · 2022-07-30
> **Response to Reviewer nC4p - Part 1**
>
> Thanks for your comments.
>
> Code and models will be released to the public.
>
> `Q-1:` Explanation of our motivation and contribution
>
> `A-1:` We highlight our novelty and technical contributions in two parts, deep unfolding framework and Transformer
>
> (a) Deep unfolding framework
>
> Our DAUF is different from previous unfolding methods in Snapshot Compressive Imaging (SCI), e.g., GAP, DNU, and ADMM. Previous methods do not estimate CASSI degradation pattern and ill-posedness degree (defined in `A-4` of `Response to Reviewer nC4p - Part 2`)
>  to guide the iterative learning. To address this issue, we formulate DAUF to estimate parameters from the compressed image and physical mask, and then use these parameters to adjust each iteration. In Tab. 2(c), DAUF outperforms DNU, ADMM, and GAP by 2.59, 1.69, and 1.63 dB. In Fig.1, DAUHST outperforms other unfolding methods by over 4 dB
>
> (b) Transformer
>
> Our HST is different from previous Transformers. In Line 68 - 71, the computational complexity of Global Transformer is enormous. The receptive fields of local Transformer are limited within position-specific windows. To tackle these issues, we propose HS-MSA to jointly develop two branches. The local branch captures local content inside patches and the non-local branch models long-range dependencies through shuffle operation. Thus, HS-MSA can enjoy larger receptive fields and relatively cheaper computational costs. In Tab. 2(b), HS-MSA outperforms G-MSA and SW-MSA by 0.42 and 0.30 dB. We provide comparisons between HS-MSA with S-MSA in `A-3` of `Response to Reviewer nC4p - Part 2`
>
> `Q-2:` Comparison with 3DT-Net [1]
>
> `A-2:` DAUHST is different from 3DT-Net in
>
> (a) Research topic
>
> 3DT-Net is proposed for Hyperspectral Image Super-Resolution (HSISR). It aims to capture a low-resolution HSI and a high-resolution multispectral image, e.g., RGB image, and fuse them into a resultant image with high spatial and spectral resolution simultaneously. This task is more like image fusion or synthesis than restoration. The main degradation of HSISR is low resolution in spatial and spectral dimensions. In contrast, DAUHST is proposed for Snapshot Compressive Imaging Reconstruction (SCIR). It aims to restore the latent 3D HSI cube from the compressed 2D measurement, see Line 34 - 38. The degradation of SCIR is modulation and compression caused by CASSI system. To the best of our knowledge, DAUHST is the first Transformer-based unfolding method for SCIR
>
> (b) Motivation and technique
>
> (i) Unfolding Framework. The unfolding framework of 3DT-Net is based on proximal gradient algorithm and observation models of LR-HSIs and HR-MSIs. 3DT-Net does not estimate degradation pattern and ill-posedness degree from input LR-HSIs and HR-MSIs to direct the iterative learning. In contrast, our unfolding framework starts from half-quadratic splitting algorithm and CASSI mathematical model, which makes the subsequent formula derivation completely different from that of [1]. Please compare Eq. (1) - (9) in [1] and Eq. (1) - (10) in our paper.  In Line 74 - 78, DAUF implicitly estimates informative parameters from the compressed measurement and physical mask. Then DAUF feeds the parameters, which capture key cues of CASSI degradation patterns and ill-posedness degree, into each iteration to adaptively scale the linear projection and provide noise level information for the denoising prior network. Besides, 3DT-Net employs some CNNs to solve the data term, e.g., the four channel or spatial operators in Eq. (6) - (9) of [1]. In contrast, we formulate a computationally efficient mathematical closed-form solution to handle data term in Eq. (10)
>
> (ii) Denoiser Prior Network. 3DT-Net adopts the Swin-Transformer layer and 3D convolution as the two key components of its prior network to exploit the spatial-spectral correlations. The adopted window-based multi-head self-attention (W-MSA) only calculates the self-attention within position-specific windows, showing limitations in capturing long-range dependencies. Besides, the prior network of 3DT-Net adopts a single-scale architecture without downsampling and upsampling the spatial resolution, which shows limitations in extracting multi-scale contextual information. In contrast, we customize a novel HS-MSA to alleviate the limitation of W-MSA. HS-MSA develops a non-local branch to model long-range dependencies through shuffle operations. In addition, HST adopts a three-level structure to capture multi-scale contextual information that is critical for HSI reconstruction. Plus, 3DT-Net shares weights across stages, limiting the representation capacity. DAUHST employs different parameters in different iterations to better fit the sophisticated SCI reconstruction
>
> We will cite and introduce 3DT-Net in revision. We are glad to conduct experiment to compare with 3DT-Net but we have not found any codes of 3DT-Net on github
>
> `Reference`
>
> [1] Learning A 3D-CNN and Transformer Prior for Hyperspectral Image Super-Resolution. Arxiv

---

> ### Comment · Area_Chair_nkAk · 2022-08-05
> **Please response to the author's rebuttal**
>
> Dear Reviewer,
>
> You concerns about the novelty and motivations are critical and can convince me to reject the paper. Please read the author's rebuttal to see if your concerns have been addressed properly.
>
> AC

---

> > ### Comment · Reviewer_nC4p · 2022-08-07
> > **Post-rebuttal discussion**
> >
> > I would like to thank the authors for addressing my comments. I don't have any concerns about the motivation of this paper now. I also have no doubt that this work has its novelty and significant contribution from the spectral compressive imaging point of view, as demonstrated by the authors' explanation and excellent experimental results. However, when the scope of consideration is enlarged to the hyperspectral image reconstruction level, the general idea of the transformer as the prior plus unfolding technique has been proposed. Although the exact task or some detailed consideration of unfolding framework and denoiser prior network is different between this paper and the prior work, the essential conceptual difference from the neural information processing point of view is not significant. Therefore, I still consider this paper as a borderline paper for NeurIPS but lean toward weak acceptance.

---

> ### Author Response · Authors · 2022-08-05
> **Further discussions with Reviewer nC4p**
>
> Dear Reviewer nC4p,
>
> We thank you for your previous review time and valuable comments. We have provided corresponding responses and results, which we believe have covered your concerns about the novelty, motivation, comparisons with 3DT-Net and MST, method description, ablation of stage number, spectral ranges, and more experiments with more infrared spectrum.
>
> We hope to further discuss with you whether your concerns have been addressed or not. If you still have any unclear parts of our work, please let us know. Thanks.
>
> Best regards,
>
> Authors

---

> > ### Comment · Reviewer_nC4p · 2022-08-08
> > **Post-rebuttal discussion**
> >
> > I would like to thank the authors for addressing my comments. I don't have any concerns about the motivation of this paper now. I also have no doubt that this work has its novelty and significant contribution from the spectral compressive imaging point of view, as demonstrated by the authors' explanation and excellent experimental results. However, when the scope of consideration is enlarged to the hyperspectral image reconstruction level, the general idea of the transformer as the prior plus unfolding technique has been proposed. Although the exact task or some detailed consideration of unfolding framework and denoiser prior network is different between this paper and the prior work, the essential conceptual difference from the neural information processing point of view is not significant. Therefore, I still consider this paper as a borderline paper for NeurIPS but lean toward weak acceptance.

---

> > > ### Author Response · Authors · 2022-08-09
> > > **Thank Reviewer nC4p for approving our work**
> > >
> > > Dear Reviewer nC4p,
> > >
> > > Thanks for discussing with us and agreeing that our response well addresses your concerns. Thank you for approving the $\textbf{motivation}$, $\textbf{novelty}$ and $\textbf{significant contribution}$ of our work for spectral compressive imaging. We will kindly cite and introduce 3DT-Net in the revision.
> > >
> > > Best regards,
> > >
> > > Authors

---

### Author Response · Authors · 2022-08-09
**Response to all reviewers and area chairs**

We thank all reviewers and area chairs for their time and valuable comments. All source code and pre-trained models will be made publicly available for further research. After discussing with reviewers and providing more clarifications/results/analyses, we would like to give a brief response.

(a) Reviewer BuNS, nC4p, and 6Fb8 all hold a $\textbf{positive}$ side for our work.

Reviewer BuNS as an expert in the field of snapshot compressive imaging highly praises the motivation, novelty, improvements, and reproducibility of our work. Reviewer nC4p has no doubt that our work has novelty and significant contribution for spectral compressive imaging. Reviewer 6Fb8 agrees with the contribution of our proposed Transformer and the practical significance of our work. Yet, reviewer 6Fb8 still has concerns about our unfolding framework. Thus, we provide further clarifications and experimental results to address the concerns of reviewer 6Fb8.

(b) Reviewer Jov1 states that our DAUHST is not better than USRNet in terms of Params and FLOPS, and our method is a combination of existing works. $\textbf{However, all these concerns have been covered in our rebuttal.}$

Firstly, in the experiments of our response, DAUHST significantly outperforms USRNet by $\textbf{2.08 dB}$ while costing $\textbf{0.24 M Params and 12.22 G FLOPS less}$ than USRNet. Secondly, we provide detailed comparisons between our DAUHST and the three works as reviewer Jov1 mentions (USRNet, CAT, and MST) in our response. They are fundamentally different in $\textbf{motivation}$, $\textbf{contribution}$, $\textbf{technique}$, etc. We also conduct comprehensive experiments to show the significant advantages of our method. Thus, we highly recommend reviewer Jov1 to read our corresponding responses carefully.

Best regards,

Authors

---

### Meta-Review · Area_Chair_nkAk · 2022-08-20

**Recommendation:** Accept
**Confidence:** Less certain

**Metareview:**

This paper integrates a Half-shuffle Transformer (HST) into the deep unfolding framework, establishing an effective method for hyperspectral image (HSI) reconstruction. The reviewers generally agree that the paper is well-written and technically-solid. The majority of the reviews assert that the technical novelty is not so dramatic. This I concur, from the perspective of learning-based reconstruction. On the other hand, from the viewpoint of spectral compressive imaging, I also agree with the authors that the paper has certain novelty and significance. Thus, I would recommend accepting the paper if the space is enough.

**Award:**

No

---

### Decision · Program_Chairs · 2022-09-14

Accept